# ORGEval: Graph-Theoretic Evaluation of LLMs in Optimization Modeling

## Abstract

Formulating optimization problems for industrial applications demands significant manual effort and domain expertise. While Large Language Models (LLMs) show promise in automating this process, evaluating their performance remains difficult due to the absence of robust metrics. Existing solver-based approaches often face inconsistency, infeasibility issues, and high computational costs. To address these issues, we propose ORGEval, a graph-theoretic evaluation framework for assessing LLMs' capabilities in formulating linear and mixed-integer linear programs. ORGEval represents optimization models as graphs, reducing equivalence detection to graph isomorphism testing. We identify and prove a sufficient condition, when the tested graphs are symmetric decomposable (SD), under which the Weisfeiler–Lehman (WL) test is guaranteed to correctly detect isomorphism. Building on this, ORGEval integrates a tailored variant of the WL-test with an SD detection algorithm to evaluate model equivalence. By focusing on structural equivalence rather than instance-level configurations, ORGEval is robust to numerical variations. Experimental results show that our method can successfully detect model equivalence and produce 100% consistent results across random data configurations, while significantly outperforming solver-based methods in runtime, especially on difficult problems. Leveraging ORGEval, we construct the Bench4Opt dataset and benchmark state-of-the-art LLMs on optimization modeling. Our results reveal that although optimization modeling remains challenging for all LLMs, DeepSeek-V3 and Claude-Opus-4 achieve the highest accuracies under direct prompting, outperforming even leading reasoning models.

## 1 Introduction

Operations Research (OR) leverages mathematical modeling and optimization techniques to support decision making in complex systems (Hillier & Lieberman, 2015). It plays a vital role across industries such as logistics, manufacturing, finance, and healthcare (Winston, 2004), where real-world scenarios are formalized into mathematical optimization models. However, this formulation process is highly challenging, demanding interdisciplinary expertise. Recently, there has been growing interest in using Large Language Models (LLMs) to automate the formulation of optimization models from user inputs, write proper programs, and solve complex problems with minimal human intervention(Jiang et al., 2024; Huang et al., 2025; Lu et al., 2025).

Evaluating modeling correctness is challenging because fundamentally equivalent optimization models can be represented by different variable names and structures. To handle this non-uniqueness, prior work has primarily adopted solver-based evaluation, which solves the generated model to obtain its objective value and then compares it against the ground-truth optimal objective (Tang et al., 2024). The assumption is that correct models should produce matching objective values. However, we noticed that solver-based evaluation suffers from a few major limitations: 1) models may coincidentally have the same optimal value under one data configuration but produce distinct values under another, 2) the solver fails to evaluate model equivalence when input data result in an infeasible problem, and 3) solvers may encounter high computation costs for a single round evaluation. Beyond optimal-value–based equivalence evaluation, Astorga et al. (2024) use satisfiability modulo theories (SMT) to assess formulation equivalence. SMT encodes formulations into logical connectives and checks their satisfiability under the feasible region of the decision variables. It is highly effective for verifying equivalence when the correspondence between variables is known. However, when the

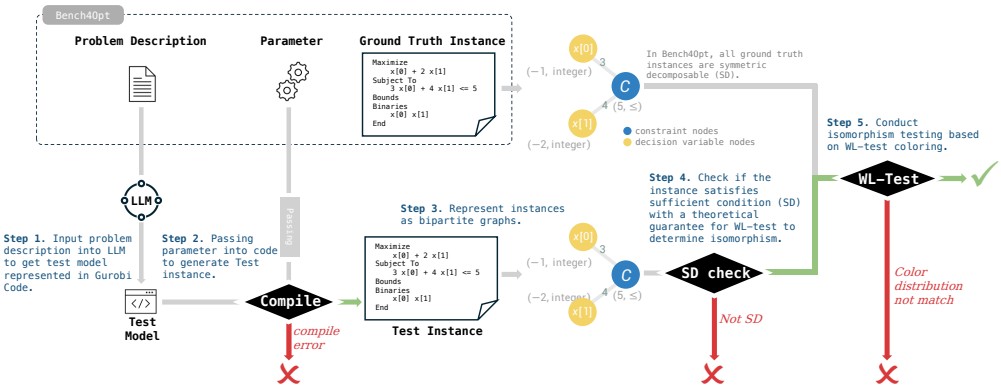

Figure 1: Evaluation Pipeline: Each example in our dataset includes a problem description, a data file, and a model instance with data applied. To assess an AI system's modeling capability, we evaluate the equivalence between the AI-generated instance and the ground truth instance in our dataset, using a common set of data. These instance pairs can be represented by two bipartite graphs, on which we applied an isomorphism testing algorithm, and meanwhile, checked the sufficiency of the algorithm.

correspondence between decision variables is not provided, SMT cannot recover this correspondence and thus fails to produce a correct evaluation. Zhai et al. (2025) attempt to prompt an LLM to infer a mapping between the decision variables of two formulations, followed by a solver-based verification step. If a valid mapping is found, the formulations are equivalent. While this offers a valuable perspective beyond objective-based evaluation, two limitations remain. First, LLM-generated mappings introduce computational overhead and may be unreliable, with hallucinated mappings or missing potential mappings. Second, the final verification still depends on a solver, which cannot reliably handle infeasible instances and may incur substantial computation when solvers struggle. Moreover, current evaluation settings focus on problems that embed both the description and the numerical data within a single prompt (Ramamonjison et al., 2022a; Xiao et al., 2023), which often constrains the problem size to relatively small instances. This approach can not reflect many real-world scenarios, where data is typically large-scale and is separated from the modeling process (ApIO et al.).

In this paper, we address these limitations by introducing ORGEval, a novel evaluation method for assessing modeling equivalence and correctness for linear and mixed-integer linear programs—the dominant classes in practical OR applications. Specifically, we formulate optimization models as graphs, then reduce the equivalence detection task to a graph isomorphism problem. We identify sufficient conditions, symmetric decomposable graphs, under which the Weisfeiler-Lehman (WL) test is guaranteed to detect isomorphism. Additionally, we develop a selection algorithm that identifies problems satisfying these conditions and combined this selection algorithm to develop an enhanced WL test that checks isomorphism.

Our main contributions can be summarized as follows:

1. We formalize model equivalence under the model–data separation setting (Section 2).

2. We reduce model equivalence detection into graph isomorphism testing (Section 3.1).

3. We identify and prove a sufficient condition that enhances existing graph isomorphism testing algorithms (Section 3.2, Section 3.3).

4. We propose ORGEval, a graph-theoretic evaluation method for assessing optimization models (Section 3.4).

5. We introduce Bench4Opt, a model–data separated dataset for optimization modeling. Using Bench4Opt, we empirically demonstrate the efficiency and consistency of ORGEval, and further benchmark the performance of leading LLMs. (Section 4)

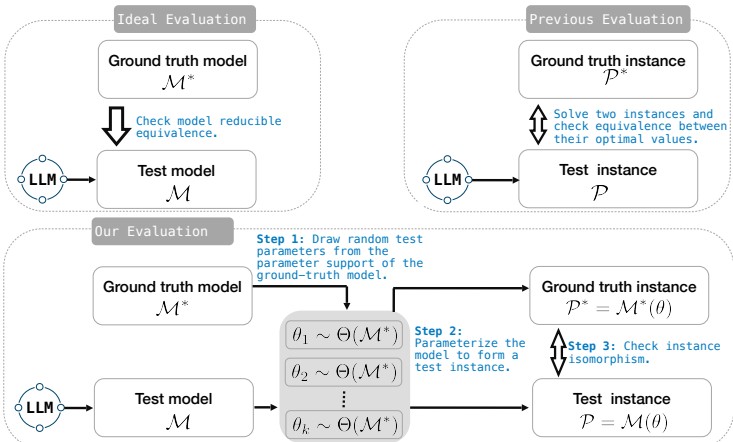

Figure 2: Evaluation Framework: The ultimate goal of modeling equivalence is to directly assess whether one model can be equivalently transformed to a standard model (top left). Existing work tests the equivalence between numerical instances by comparing their optimal objective (top right). Our evaluation method approximates the ultimate goal of directly evaluating modeling equivalence by randomly sampling instances and testing instance isomorphism. (bottom)

## 2 PROBLEM FORMULATION

In this section, we establish the background for the OR modeling task and define the capabilities we aim to measure and evaluate. First, we note that real-world OR modeling typically unfolds through three distinct stages:

- **Pre-modeling stage:** Stakeholders articulate problems in natural language (e.g., "I want to maximize revenue through car production") and collect all relevant numerical parameters.

- **Modeling stage:** Analysts transform these natural language descriptions into mathematical models by defining decision variables, formulating objective functions, and establishing constraints. This results in mathematical formulations or solver-ready code with separately stored parameters.

- **Post-modeling stage:** The model is instantiated with problem-specific data to generate a fully specified problem ready for computational solving.

We explicitly distinguish between the modeling stage and the post-modeling stage. In many real-world applications, models are designed for reusability: the structural components (decision variables, objectives, and constraints) are specified once, while the problem-specific parameters are supplied separately for each instance. This separation is standard practice in modern modeling languages such as AMPL (Fourer et al., 2003) or Pyomo (Hart et al., 2011), where the model (.mod) and the data (.dat) are maintained as distinct artifacts. We thus view "instantiation with problem-specific data" as a necessary step that bridges abstract modeling and computational solving.

Our primary goal is to leverage LLMs to automate the first two stages, enabling users to input problem descriptions and receive both mathematical formalizations and solver-ready code as outputs. We formally reframed the evaluation of modeling equivalence in the following subsections and illustrated ideal evaluation, previous evaluation, and our new evaluation framework in Figure 2.

### 2.1 DEFINITION OF MODELING "ACCURACY"

Our work proposes a formal evaluation framework for model-level "accuracy", which requires the definition of model equivalence. To support the definition of this kind of equivalence, we introduce the notions of *modeling problem instance* and *modeling parameter support*.

### 2.1.1 Definitions for Modeling Problems

**Definition 1** (Modeling Problem Instance). *A MILP/LP problem instance, denoted by $\mathcal{P}$, has the following standard formulation ([Luenberger et al., 1984](#)):*

$$\min_{\mathbf{x} \in \mathbb{R}^p \times \mathbb{Z}^{n-p}} \mathbf{c}^\mathsf{T}\mathbf{x}, \tag{1}$$

$$\text{such that} \quad \mathbf{A}\mathbf{x} \circ \mathbf{b},$$

*where $\mathbf{A} \in \mathbb{R}^{m \times n}$, $\mathbf{c} \in \mathbb{R}^n$, $\mathbf{b} \in \mathbb{R}^m$, and $\circ \in \{=, <, >, \leq, \geq\}^m$ for $i = 1, \cdots, n$. Note that in this formulation, there are $p$ real optimization variables and $n - p$ integer optimization variables, and $m$ constraints.*

To facilitate later usage, we use a vector $\tau$ to represent the decision variable type, where $\tau_i = 1$ indicates $x_i$ is an integer and $\tau_i = 0$ indicates a continuous variable.

**Definition 2** (MILP/LP Model). *A MILP/LP model is a mapping $\mathcal{M} : \Theta \to \mathcal{P}_{A,b,c,\circ,\tau}$, where $\Theta$ denotes a space of problem data and $\mathcal{P}$ is a model instance with parameters $(A, b, c, \circ, \tau)$. Here, $\Theta$ may be any subset of $\mathbb{R}^q$ that represents the admissible "data" of the modeling family, where $q$ is a dimension of problem data.*

In practice, part of $\Theta$ is fixed (as structural components), and the remaining part consists of real-valued elements that vary within a closed subset of $\mathbb{R}$. We refer to $\Theta$ as the **problem data support** of $\mathcal{M}$. Given any $\theta \in \Theta$, the mapping $\mathcal{M}(\theta)$ returns a concrete MILP or LP instance $\mathcal{P}$ of the form Eq. (1). An example of a model and its problem data support can be found in Example 4.

### 2.1.2 Definitions for Modeling Equivalence

In practical optimization workflows, "modeling equivalence" should reflect whether one predicted model can be systematically transformed to a target model.

**Definition 3** (Model-lossless-reduction). *For two models $\mathcal{M}_1$ and $\mathcal{M}_2$, they are said to be model-lossless-reducible if the following conditions hold:*

1. *Shared problem data support: $\mathcal{M}_1$ and $\mathcal{M}_2$ share the same problem data support $\Theta$;*
2. *Existence of solution-preserving transformation: There exists a mapping $F$ over decision variables such that, for any problem data $\theta \in \Theta$:*
   - *If $\mathcal{M}_1(\theta)$ is feasible and bounded, then $\mathcal{M}_2(\theta)$ is also feasible and bounded, and for any optimal solution $x^*$ of $\mathcal{M}_1(\theta)$, $F(x^*)$ is an optimal solution of $\mathcal{M}_2(\theta)$;*
   - *If $\mathcal{M}_1(\theta)$ is infeasible or unbounded, then $\mathcal{M}_2(\theta)$ is also infeasible or unbounded.*

This model-lossless-reduction captures the essence of equivalence between models: if one model can fully simulate the feasible behaviors and optimal outcomes of another, we treat them as equivalent for all practical purposes.

However, such a form of equivalence is difficult to check and has not been reliably captured by existing works. First, it is hard to verify the equivalence between parameter support. Second, verifying solution mappings $F$ between models can be computationally expensive and sometimes impractical, as it may require comparing optimality across a large space of instances rather than a single solution.

As verifying model-lossless-reducibility is often intractable in practice, current approaches typically rely on a much weaker proxy, comparing solver outputs on specific model instances. In the following section, we will formally introduce solver-based modeling accuracy evaluation and discuss its limitations.

## 2.2 Solver-based Modeling "Accuracy"

Previous works ([Tang et al., 2024](#)) explored evaluating modeling "accuracy" through execution accuracy, which we formalize below:

**Definition 4** (Execution Accuracy). *For a data configuration $\theta$, a mathematical model $\mathcal{M}$ with a program code $\mathcal{C}$ is said to be correct if, upon execution of $\mathcal{C}$, we obtain $z(\mathcal{C}, \theta) = z^\star$, where $z^\star$ is the optimal value of the mathematical model if it exists.*

We point out several limitations of this definition.

- **(Limitation 1) No rigorous correctness guarantee when the solver returns values:** Cases exist where final answers appear correct despite fundamentally flawed underlying optimization models; see example 1 and example 2 in appendix E.

- **(Limitation 2) Uninformative in cases of infeasible problems** Mathematical models could be *infeasible* under certain data configurations, in which cases the solver would return no feasible solution. Solver-based evaluation becomes uninformative in this scenario; see 3 in appendix E.

- **(Limitation 3) Prohibitively high computational costs:** Solvers may require hundreds and thousands of CPUs and run for several hours and days, especially for large-scale problems such as Mixed Integer Linear Programming (MILP) tasks. This makes the evaluation time-consuming and computationally expensive. This would further make it impractical to apply advanced post-training techniques like those proposed by (OpenAI, 2024; Guo et al., 2025) that could otherwise enhance LLM performance.

Fundamentally, the limitations illustrated above reveal a critical limitation in the execution accuracy definition: it is restricted to a *single* data configuration and relies solely on optimal value comparison. This approach is clearly inadequate for reliable model evaluation. A mathematical model should be correct across *all* possible data configurations, not just one specific test case. Only when a model demonstrates consistent performance across diverse scenarios can we trust the decision information it provides for practical applications. Therefore, an ideal evaluation metric for model equivalence is expected to be reliable, informative, and consistent over various data configurations.

### 2.3 MODEL ISOMORPHISM

As an alternative, we further define model isomorphism to capture structural equivalence between models, rather than relying on numerical optimal values.

**Definition 5** (Model Isomorphism). *We say two optimization models $\mathcal{M}_1$ and $\mathcal{M}_2$ are model isomorphic if the following conditions hold:*

- *Shared problem data support: $\mathcal{M}_1$ and $\mathcal{M}_2$ share the same problem data support $\Theta$;*

- *Existence of permutation-invariant transformation: There exists a permutation mapping $F_1$ over decision variables and $F_2$ over constraints such that, for any $\theta \in \Theta$, the transformed instance $F_1 \circ F_2(\mathcal{M}_1(\theta))$ is exactly $\mathcal{M}_2(\theta)$.*

In this work, we refer to this as isomorphism equivalence, or simply equivalence. Note that model isomorphism is sufficient for mutual reducibility: if $\mathcal{M}_1$ and $\mathcal{M}_2$ are isomorphic, then $\mathcal{M}_1$ and $\mathcal{M}_2$ are mutually model-lossless-reducible.

**Optimization Model Equivalence**  Previous work typically assesses instance-level equivalence by comparing the optimal values of $\mathcal{P}_1 = \mathcal{M}_1(\theta)$ and $\mathcal{P}_2 = \mathcal{M}_2(\theta)$ for specific $\theta$, where both the model and data are coupled in the input prompt. Different from previous work, we aim to detect model-level equivalence and consider a model-data separated framework for evaluating autonomous modeling systems. To make model-level equivalence evaluation tractable, we reduce model "equivalence" to model isomorphism, a stricter form of equivalence that focuses on the inherent structure of models rather than their optimal solutions. Importantly, we observe that if two instances are isomorphic, their isomorphism remains unchanged under different data configurations. This property allows us to efficiently evaluate equivalence at the model level without repeatedly solving individual model instances.

Specifically, we evaluate equivalence between $\mathcal{M}_1$ and $\mathcal{M}_2$ by randomly sampling problem data $\theta$ from $\Theta(\mathcal{M})$ and testing the isomorphism of modeling instances $\mathcal{M}_1(\theta)$ and $\mathcal{M}_2(\theta)$. This serves as an empirical approximation of model-level correctness.

## 3 METHODOLOGY

In this section, we introduce our evaluation method ORGEval, and the theoretical guarantee of ORGEval. ORGEval is developed to evaluate modeling equivalence by detecting whether the inherent structure of two random instances from two models is equivalent, thus making the evaluation stable when altering instance problem data.

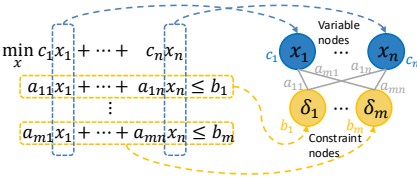

Figure 3: Transform model instance to a bipartite graph.

### 3.1 EVALUATION PRINCIPLE

We first introduce our notion of **equivalence** between two model instances. Specifically, our notion allows model instances to change the notations of variables and rearrange their variables/constraints in different orders without losing essential information. The formal definition is as follows:

**Definition 6** (Instance-Level Isomorphism). *We say model instances $\mathcal{P}_2$ and $\mathcal{P}_1$ are isomorphic, if $\exists$ permutation matrices $P_1, P_2$ which shuffle the index of a vector or column index of a matrix such that $\mathcal{P}_2$ can be written in the following form:*

$$\min_{\mathbf{x} \in \mathbb{R}^{n-p} \times \mathbb{Z}^p} \hat{\mathbf{c}}^T \mathbf{x},$$

$$s.t. \ \hat{\mathbf{A}}\mathbf{x} \, \hat{\circ} \, \hat{\mathbf{b}},$$

*where $\hat{\mathbf{b}} = P_2\mathbf{b}, \hat{\mathbf{c}} = P_1\mathbf{c}, \hat{\mathbf{A}} = P_2\mathbf{A}P_1, \hat{\circ} = P_2\circ$.*

We denote that two instances $\mathcal{P}_1$ and $\mathcal{P}_2$ are equivalent or instance-level-isomorphic by $\mathcal{P}_1 \sim \mathcal{P}_2$. Note that we require the model formulation to strictly adhere to the textual description of the problem and account for differences in formulation strength. For example, adding slack variables to the standard instance may not change the optimal solution, but it alters the direct physical meaning specified in the textual description. Therefore, we consider it a different formulation. Consequently, our definition of model equivalence only allows altering the variable notations and permutating variables/constraints, which is in general stricter than that conventionally adopted in OR.

Our concept of model equivalence aligns with the isomorphism of graphs, allowing nodes to be re-indexed or rearranged without changing the graph structure. This motivates us to incorporate tools in graph theory to evaluate model equivalence. Following existing work in learning to optimize (Gasse et al., 2019; Chen et al., 2022b), we represent an LP/MILP model instance by a bipartite graph (see Figure 3 for an illustrative example). We proved that detecting the model equivalence can be reduced to testing graph isomorphism; See Appendix D.3 for a formal demonstration.

### 3.2 MODEL EQUIVALENCE BASED ON GRAPH ISOMORPHISM

We evaluate the modeling result in two steps:

**Create test and standard graphs**  We can use (weighted) bipartite graphs to equivalently represent the modeling problem instance. We follow the formal notation from Chen et al. (2022b) to represent (MI)LP instances to bipartite graphs as follows:

**Definition 7** (Weighted Bipartite Graph Instance Representation). *A MILP/LP instance can be represented as a weighted bipartite graph $\mathbf{G} = (\mathbf{V} \cup \mathbf{W}, \mathbf{E})$, where $\mathbf{V} = \{\mathbf{v}_1, \ldots, \mathbf{v}_m\}$ corresponds to constraints and $\mathbf{W} = \{\mathbf{w}_1, \ldots, \mathbf{w}_n\}$ corresponds to variables. Each edge $(v_i, w_j) \in \mathbf{E}$ is weighted by the coefficient $\mathbf{A}_{ij}$, and each vertex is associated with features (e.g., right-hand side $b_i$, operator type $\circ_i$ for constraints; objective coefficient $c_j$, integrality type $\tau_j$ for variables).*

Since its dependence on $\theta = (\mathbf{A}, \mathbf{c}, \tau, \mathbf{b}, \circ)$, we may write $\mathbf{G}(\theta)$. See Figure 3 for an example to transform a modeling problem instance into a bipartite graph instance.

As introduced in Definition 7 and Figure 3, we represent MILP/LP instances as bipartite graphs. In such graphs, nodes can be divided into two groups: variable nodes and constraint nodes. All nodes are equipped with the necessary features. Each constraint node connects with all associated variable nodes. Given such graph representations, we can use graph isomorphism testing tools to detect equivalence between models.

**Isomorphism testing** Graph isomorphism testing is a challenging problem, with no known polynomial-time algorithm to date (Garey & Johnson, 1979; Babai, 2016). The Weisfeiler-Lehman test (Leman & Weisfeiler, 1968) is an effective and computationally efficient approximation for graph isomorphism testing. Typically, one may determine that two graphs are non-isomorphic if the WL-test algorithm produces different outputs (in the form of so-called "coloring distributions"). However, if the WL-test yields the same outputs for the two graphs, they are not guaranteed to be isomorphic (Cai et al., 1992); See Appendix E for counterexamples.

In contrast to the widely used WL-test, which offers no theoretical guarantee of producing correct results within polynomial time, our enhanced algorithm first verifies the satisfaction of a sufficient condition. This additional step establishes a formal guarantee of correctness for the subsequent equivalence evaluation.

### 3.3 SUFFICIENT CONDITION FOR GRAPH ISOMORPHISM TESTING

We propose a sufficient condition, say symmetric decomposable, that modeling instances should satisfy to be testable by a polynomial-time isomorphism testing algorithm.

**Definition 3.1** (Symmetric Decomposable Instance)**.** We say a modeling instance $\mathcal{P}$ is **symmetric decomposable** if, after WL-test, the coloring on its representation graph $\mathcal{G}$ satisfies the following conditions: Excluding nodes that are uniquely colored, the remaining nodes can be divided into $k$ disjoint groups (with some $k \geq 0$) of the same size, denoted by $S_1, S_2, \cdots, S_k$, where

1. All nodes in the same group have distinct colors,
2. All groups share the same coloring sets, and
3. Every two groups are disconnected, i.e. $\forall$ nodes $a \in S_i, b \in S_j, i \neq j$, $a$ is disconnected with $b$.

In previous work, Chen et al. (2022b) characterized one class of graph conditions, termed unfoldable graphs, whose isomorphism can be accurately distinguished by WL-Test. Yet the underlying graphs of many MILP problems do not fall into this category. For example, graphs for bin-packing instances are typically not unfoldable; see example 7. The symmetric decomposable condition broadens the scope of problems for which WL-based isomorphism detection is guaranteed.

While the unfoldable property requires all nodes to have distinct colors, the symmetric decomposable property is more relaxed. It allows the graph to be partitioned into several subgraphs such that within each subgraph, every node has a distinct color. Note that when $k = 1$, a symmetric decomposable problem is reduced to being unfoldable. Example of a decomposable instance can be found in Figure 12, Figure 13, and Figure 14.

In the following theorem, we show that if the standard instance is symmetric decomposable, then Algorithm 1 is reliable for detecting whether a test instance is model-equivalent to the standard instance. Rigorous proof can be found in Appendix D.6.

**Theorem 3.1.** *Suppose* $\mathcal{P}_1, \mathcal{P}_2$ *are symmetric decomposable, then* $\mathcal{G}_1$ *and* $\mathcal{G}_2$ *shares the same coloring distribution after WL-test coloring* $\iff \mathcal{P}_1 \sim \mathcal{P}_2$.

To leverage theorem 3.1 in practice, we designed an algorithm to identify symmetric decomposable instances (Algorithm 3). Moreover, we prove that under mild assumptions, a random sample yields a symmetric decomposable instance with high probability (Theorem D.5, D.6). Leveraging this property, we construct a benchmark dataset in which all ground-truth instances are guaranteed to be symmetric decomposable.

### 3.4 ORGEVAL: MODEL-EQUIVALENCE EVALUATION BASED ON GRAPH

Combined with the symmetric decomposable detection algorithm (Algorithm 3), we develop ORGEval, a variant form of WL-test to test model equivalence. ORGEval can accurately detect whether a test instance is equivalent to a symmetric decomposable ground truth. It involves 1) running a WL-test for the LP/MILP test and standard instances[1]; 2) checking whether the test instance is symmetric decomposable based on its coloring distributions (done by Algorithm 3). If not, since all ground truth instances are selected to be symmetric decomposable, the test instance must be different from a ground truth one, so the algorithm returns "not equivalent". If yes, go to step 3. 3) checking

---

[1]We use the same implementation as Chen et al. (2022b), presented as Algorithm 2.

whether the two coloring distributions are identical. If yes, then return "equivalent"; if not, then return "not equivalent".

**Efficiency of ORGEval** The time complexity to distinguish tested problem instances from the standard instances with $m$ variables and $n$ nodes is at most $\mathcal{O}(kmn)$, where $k$ is the number of clusters in a symmetric decomposable graph. This is far better than the complexity for exhaustive isomorphism testing; detailed complexity analysis can be found in Appendix D.8.

---

**Algorithm 1** Modeling Equivalence Detection

---

**Require:** Two graph instances $(G_k, H_k) \in \mathcal{G}_{m,n}^k \times \mathcal{H}_m^V \times \mathcal{H}_n^W$ and adjacency matrix $\mathbf{A}_k, k = 1, 2$; iterate limit $L > 0$.
1: Color nodes in two graphs using WL-test Algorithm for MILP/LP, get two coloring multi-sets $\mathcal{C}_k = \left\{ \{\{C_i^{k,V}\}\}_{i=0}^m, \{\{C_j^{k,W}\}\}_{j=0}^n \right\}, k = 1, 2$ for coloring $\mathcal{G}_1$ and $\mathcal{G}_2$.
2: Derive set of unique elements in $\mathcal{C}_k$, denote as set $\mathbb{A}_k, \forall k = 1, 2$.
3: **if** $\mathcal{C}_1 \neq \mathcal{C}_2$ **then**
4:     **return** Not equivalent
5: **else if** $\mathcal{G}_2$ are symmetric decomposable **then**
6:     **return** Equivalent
7: **else**
8:     **return** Not Equivalent

---

## 4 EXPERIMENT AND ANALYSIS

We present comprehensive experiments on ORGEval using Bench4Opt, a benchmark dataset for optimization modeling that separates models from data. Bench4Opt comprises 197 LP and MILP instances curated from both hand-crafted problems and the MIPLIB dataset (Gleixner et al., 2021). We empirically demonstrate the efficiency and consistency of ORGEval in Section 4.2, and further employ it to benchmark the performance of LLMs in optimization modeling in Section 4.3.

### 4.1 BENCHMARK DATASET

To robustly assess ORGEval, we introduce Bench4Opt, a diverse benchmark dataset consisting of 394 optimization modeling word problems in a model-data separated format. Bench4Opt encompasses multi-dimensional complexity through a hierarchical reverse data evolution, including optimization problem **type**, **classes**, **domains**, **variants**, and **level of abstraction**. Each optimization model is represented in 2 levels of description abstraction. Besides careful verification and quality control, we applied our sufficient condition detection Algorithm 3 to evaluate the problem instances in our benchmark dataset. While we did not intentionally select problem instances based on specific criteria, we found that all problems in Bench4Opt satisfy our sufficient conditions. An illustrative example from Bench4Opt can be found in Figure 6 and Figure 5. Detailed construction pipeline and data coverage can be found in Section B

### 4.2 ADVANTAGES OF ORGEVAL

**Evaluation Efficiency** Empirically, we demonstrate that ORGEval offers significantly higher evaluation efficiency compared with solver-based methods, especially for problem instances that are challenging for solvers. To test this, we sampled 75 problem instances from the MIPLIB (Gleixner et al., 2021) dataset across three difficulty levels: easy, hard, and open for existing solvers, with 25 instances per level. According to MIPLIB's definition, easy instances can be solved within an hour using a standard solver on a typical desktop machine with up to 16 threads; hard instances require a longer time, non-standard hardware, or advanced algorithms; and open instances have not yet been solved. Solver often requires hours or more to evaluate selected instances from MIPLIB. In contrast, ORGEval consistently produced evaluation results within a reasonable timeframe— ORGEval runs 30 seconds on average to output an evaluation result, even for the most challenging open problems for solvers. See Table 1 for the average evaluation time of ORGEval across the three difficulty levels.

Table 1: Evaluation time of ORGEval for three levels of difficulties: easy, hard, open. Instances are sampled from MIPLIB, with 25 instances per level.

| Level of Difficulty | Average Problem Size (#variables + # constraints) | Average Evaluation Time (Solver) | Average Evaluation Time (ORGEval) |
|---|---|---|---|
| Easy | 1848 | about 1 hour | **0.21s** |
| Hard | 10463 | > 1 hour | **3.83s** |
| Open | 17050 | not yet being solved | **32.07s** |

**Evaluation Consistency** We use the evaluation result of five random instances to indicate equivalence between two models. For such an evaluation to be valid, we address the consistency of our evaluation result across various data configurations: Our experimental results show that ORGEval achieves 100% consistency across five random data configurations for all models in Bench4Opt. In contrast, as illustrated in table 2, solvers fail to evaluate models with random data configurations for more than 60% of the models due to infeasibility, and even among the remaining models, 5.89% of model pairs yield inconsistent results under solver-based equivalence evaluation.

Table 2: Comparison of model consistency across 5 random instances under different evaluation schemes. For each Bench4OPT problem, we evaluate LLM-generated formulations using both the Solver-based method and ORGEval over five randomly instantiated data instances. The table reports the proportion of instances for which the two evaluation schemes yield consistent results. Feasibility consistency denotes the fraction of problems whose five instances can all be solved by the solver.

| Model | Feasibility consistency | ORGEval consistency | Solver consistency |
|---|---|---|---|
| gpt-4o | 36.30% | 100.00% | 95.58% |
| claude-opus-4 | 36.05% | 100.00% | 92.12% |
| deepseek-v3 | 34.69% | 100.00% | 93.95% |
| o1 | 35.43% | 100.00% | 94.77% |
| **Average** | 35.62% | 100.00% | 94.11% |

## 4.3 BENCHMARK THE MODELING ABILITY USING ORGEVAL AND BENCH4OPT

To assess the capabilities of LLMs in optimization modeling, we conducted a comprehensive evaluation using the Bench4Opt benchmark. Our evaluation focused on top-performing LLMs via direct prompting. The main evaluation result is listed in Table 3.

Table 3: Evaluation Results on Bench4Opt. We report pass@1 performance for each LLM. The Overall column summarizes results across the full Bench4Opt suite. The Breakdown column reports performance at two levels of abstraction—structured and unstructured. Examples of each problem type are provided in Figure 6 and Figure 5. SOTA scores within each category are highlighted in red.

| LLMs | Overall | | Breakdown | |
|---|---|---|---|---|
| | Accuracy% | Compile Error% | Structured% | Unstructured% |
| deepseek-v3 | 54.82 | 2.28 | 63.45 | 46.19 |
| claude-opus-4 | 54.82 | 2.28 | 60.41 | 49.24 |
| gpt-4.1 | 52.28 | 1.78 | 57.36 | 47.21 |
| gpt-4o | 51.02 | 7.36 | 58.38 | 43.65 |
| claude-opus-4.1 | 50.76 | 0.76 | 59.39 | 42.13 |
| o3 | 47.97 | 0.76 | 55.84 | 40.10 |
| deepseek-r1 | 47.72 | 2.28 | 55.84 | 39.59 |
| o1 | 47.21 | 1.78 | 52.79 | 41.62 |

Our benchmark revealed varying performances among the tested models. In particular, claude-opus-4 (Anthropic, 2025) and deepseek-v3 (Liu et al., 2024a) achieved the strongest results, each reaching an overall accuracy of 54.82%, outperforming other contenders across structured and unstructured optimization tasks. In contrast, reasoning models such as deepseek-r1 (Liu et al., 2024a), o1 (OpenAI, 2024), and o3 (OpenAI, 2025)consistently exhibited lower accuracy compared to the base model. While reasoning models produce outputs with relatively fewer compile errors, their multi-step reasoning capabilities appear susceptible to hallucination propagation. The cascading effect of these reasoning artifacts likely contributes to progressive accuracy degradation throughout complex problem-solving sequences. This phenomenon may explain the observed performance gap despite their enhanced error-handling capabilities.

## 5 CONCLUSION

In this work, we formalize the task of detecting equivalence between (MI)LP models and introduced a new modeling equivalence evaluation method, **ORGEval**, to evaluate the equivalence between optimization models. By representing optimization models as graphs and leveraging the Weisfeiler-Lehman (WL) test under well-defined sufficient conditions, our method offers a principled and efficient alternative to solver-based evaluations. Our experiments demonstrate that ORGEval achieves consistent evaluation results across all data configurations and offers significant speed advantages over solver-based methods, particularly on hard-to-solve problems. To access ORGEval , we also introduce Bench4Opt, a diverse benchmark dataset of 394 model-data-separated optimization problems containing problem from MIPLIB dataset and hand crafted problem generated with the help of LLM.

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

## A  LLM USAGE STATEMENT

In this work, we used large language models (LLMs) in two ways:

**Paper Revision:**  LLMs were used to assist in improving the clarity and readability of the manuscript. All scientific content, results, and interpretations were reviewed, verified, and rewritten by the authors.

**Benchmark Construction:**  LLMs played a role in constructing and expanding the benchmark dataset. Specifically, they assisted in generating candidate problem instances and verifying problem formulations. All final benchmark data and evaluations were curated and validated by the authors to ensure correctness and consistency.

## B  DATASET

### B.1  DATA CONSTRUCTION

To efficiently construct multidimensional complexity, we employ a hierarchical reverse data evolution pipeline comprising three key stages: optimization model stimulation, reference answer generation, and word problem generation, see Figure 9.

**Optimization Models Stimulation**  Our dataset is constructed from two categories of seed problems: 1) seed problems proposed by an LLM, and 2) seed problems selected from MIPLIB. For the LLM-proposed seed problems, we employ GPT-4o to systematically generate optimization models by progressively expanding the problem context across types, classes, domains, and variants. GPT-4o first identifies classical problem classes within linear programming (LP) and mixed-integer linear programming (MILP). And each seed For each class, it then selects representative application domains to ensure practical relevance. Within each domain–problem-class pair, GPT-4o produces a canonical optimization model and subsequently enriches it by varying core modeling components—objectives, constraints, and decision variables—to introduce structured diversity and increased complexity. In addition, we incorporate seed problems from the MIPLIB dataset. Operations research expert translates the selected MIPLIB instances into abstracted Gurobi model code along with the necessary data files. Because directly converting from the original LP files is often challenging, we restrict our selection to relatively small instances; the largest contains 48 variables and 34 constraints. We then use GPT-4o to rewrite these MIPLIB-derived models by generating domain-specific adaptations and variant formulations.

**Reference Answers Generation**  For problems augmented from MIPLIB seed problem, we ask domain experts to annotate the corresponding data-generation code, enabling multiple generations of associated parameter files (parameter.json) . For problems augmented from LLM-generated seed instances, we used GPT-4o to automatically produce Python programs tailored to each optimization model and to generate the required parameter.json files. Each simulated optimization model then reads its associated parameter file and produces the corresponding .lp instance (model.lp). This procedure could also be helpful if variation in problem size is needed by modifying dimensionality specifications directly in the prompts, thereby streamlining the expansion of individual optimization models.

**Word Problems Generation**  Finally, we reversely generate word problems from the stimulated optimization models. Drawing inspiration from the INFORMS AIMMS-MOPTA AIMMS (2024) Optimization Modeling Competition, we meticulously designed a standardized word problem structure, as illustrated in Figure 8. Using GPT-4o, we first translate the solver code into detailed word problems adhering to this standardized structure. Subsequently, we refine and summarize the generated content to produce concise and unstructured problem statements. This approach ensures that the generated word problems accurately represent the underlying optimization models while being suitable for benchmarking in a complex and realistic manner.

## B.2 QUALITY CONTROL

To pursue high data quality in our benchmark, we employ a controlled generation framework with systematic verification.

**Controlled Generation** Similar to the design of standardized word problem structures, we develop a structured code skeleton tailored to guide optimization models. This approach transforms an open-ended generation task into a constrained code completion task, enabling tight regulation of LLM output. By partially automating the data evolution and verification pipeline, our framework ensures consistency across generated instances and adherence to predefined structural requirements, as illustrated in Figure 7.

**Verification** We systematically validate each model-generated answer pair. During model simulation and answer generation, we execute the corresponding code representations, retaining only those that run without errors. Additionally, in the word problem generation stage, we leverage an LLM-based verifier to confirm the precise alignment between optimization components in the code and their textual descriptions. To further enhance reliability, we involved 4 PhD students with expertise in OR in the verification The verification consists of two stages:

- **Alignment Verification** To ensure a strict alignment between each word problem and its associated model code. Specifically, all objectives, decision variables, and constraints described in the word problem must be accurately reflected in the model code, with no omissions or extraneous elements. All enerated problems were reviewed by the experts. Minor issues (e.g., typographical errors or minor inconsistencies) were corrected. Problems exhibiting major mismatches—such as missing or incorrect objectives, variables, or constraints—were discarded.

- **Representative Verification** To verify that the reference model codes align with the word problems. A random sample of 50 word problems was selected. Experts constructed their own optimization model based on the word problem descriptions, without reference to the original code. The resulting models were then compared to the reference implementations. The experts' answers for all 50 word problems match the reference answer.

By integrating controlled generation with rigorous verification, we uphold high standards of data quality and accuracy. This ensures the reliability and robustness of our benchmark for evaluating large language models in operations research modeling.

## B.3 DATASET COVERAGE

| Problem Types | Problem Classes |
|---|---|
| LPs | Diet Problem |
| | Transportation Problem |
| | Blending Problem |
| | Production Planning Problem |
| | Network Flow Problem |
| | Portfolio Optimization Problem |
| | Cutting Stock Problem |
| | Staff Scheduling Problem' |
| MILPs | Knapsack Problem |
| | Traveling Salesman Problem (TSP) |
| | Vehicle Routing Problem (VRP) |
| | Bin Packing Problem |
| | Set Covering Problem |
| | Capacitated Facility Location Problem |
| | Capital Budgeting Problem |
| | Assignment Problem |

Table 4: Optimization problem types and classes covered in our Bench4Opt.

| Category | Representative Application Scenarios |
|---|---|
| **Transportation, Routing & Logistics** | Vehicle routing, cargo/vehicle loading, postal and parcel delivery, food delivery, waste collection, public transportation planning, school bus routing, logistics and distribution, transportation network design. |
| **Supply Chain, Production & Manufacturing** | Supply chain management, production planning, manufacturing scheduling, agricultural/food/pharmaceutical distribution, industrial material flow, chemical/metal/glass/paper/textile manufacturing. |
| **Facility Location & Spatial Planning** | Warehouse location optimization, facility location and assignment, fire station placement, healthcare facility location, wildlife reserve planning, public infrastructure placement, districting and territory design. |
| **Workforce, Scheduling & Operational Planning** | Employee rostering, hospital nurse scheduling, call center scheduling, university professor scheduling, public transport driver shifts, construction worker scheduling, classroom scheduling, school timetabling, project task assignment. |
| **Resource Allocation, Budgeting & Decision Optimization** | Portfolio optimization, investment planning, advertising budget allocation, general resource allocation, R&D portfolio planning, energy distribution and generation, telecommunications network design and frequency allocation. |
| **Agriculture, Environment & Public Services** | Agricultural land use planning, animal feed formulation, fertilizer optimization, environmental conservation, water and wastewater management, emergency services deployment, healthcare services, nutrition planning for hospitals, schools, and elderly care. |

Table 5: Industry domain covered in our Bench4Opt.

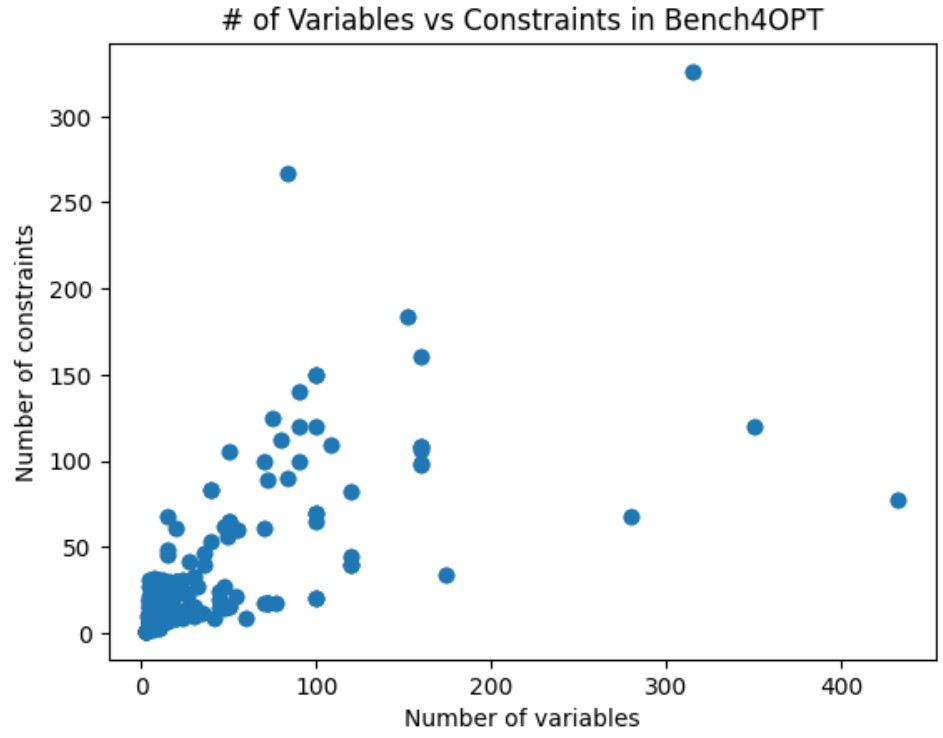

Figure 4: Optimization problem size covered in our Bench4Opt.

```
In a cargo loading scenario, you need to choose a subset of items, each with a given value
and weight, to maximize total value without surpassing the vehicle's weight capacity. The
decision to include an item is binary. You'll be given a list of item values, weights, and
the vehicle's capacity. Your task is to determine which items to include to achieve the
highest total value while staying within the weight limit.

You should only consider parameters listed below. And these parameters will be provided in a
separated "data.json".
{
  'values': 'the value of each item; list of length (number of items)',
  'weights': 'the weight of each item; list of length (number of items)',
  'capacity': 'the capacity of the vehicle; single float value',
}
```

Figure 5: Example for concise version word problem on cargo loading.

```
**Problem Statement: Knapsack Problem in Cargo Loading**

**Background:**
In the context of cargo loading, the knapsack problem involves selecting a subset of items
to include in a cargo such that the total value of the selected items is maximized, while
ensuring that the total weight of the selected items does not exceed the vehicle's capacity.
This problem is a classical example of a combinatorial optimization problem and is widely
studied in operations research.

**Problem Description:**
Given a set of items, each with a specific value and weight, the objective is to determine
which items to include in the cargo to maximize the total value without exceeding the
vehicle's weight capacity. The decision to include an item in the cargo is binary (either
the item is included or it is not).

**Parameters:**
Only consider parameters listed below. And these parameters will be provided in a separated
"data.json".
{
   'values': 'the value of each item; list of length (number of items)',
   'weights': 'the weight of each item; list of length (number of items)',
   'capacity': 'the capacity of the vehicle; single float value',
}

**Decision Variables:**
- \( x[i] \): A binary variable that indicates whether item \( i \) is included in the cargo
(1) or not (0).

**Objective:**
Maximize the total value of the items included in the cargo. This is achieved by summing the
product of the value of each item and its corresponding binary decision variable.

**Constraints:**
The total weight of the items included in the cargo cannot exceed the vehicle's capacity.
This is ensured by summing the product of the weight of each item and its corresponding
binary decision variable and ensuring that this sum does not exceed the given capacity.

**Implementation Notes:**
- The problem is formulated as a Mixed-Integer Linear Programming (MILP) problem.
- The decision variables are binary, indicating the inclusion or exclusion of each item.
- The model should be saved as a '.lp' file for further analysis and solution.

**Expected Outcome:**
The expected outcome is a selection of items that maximizes the total value while ensuring
that the total weight does not exceed the vehicle's capacity. The solution will provide the
optimal set of items to include in the cargo.
```

Figure 6: Example for word problem on cargo loading.

```python
import json
from gurobipy import Model, GRB

# {problem_class} — {problem_name}
# Problem type: {problem_type}
# Domain: {domain}
# Property: (fill in this comment by briefly describing the
variant of the problem)

# Read data
with open('data.json', 'r') as f:
data = json.load(f)

### (fill in this section) Read parameters from data (assign
domain specific parameter name)

### (fill in this section) Get hyperparameter from parameters
(assign domain specific parameter name)

# Create a new model
model = Model("{problem_class}")

### (fill in this section) Add variables of the classic
{problem_name} (assign domain specific name)

### (fill in this section) Set objective of the {problem_name}
(assign domain specific name)

### (fill in this section) Add constraints of the
{problem_name} (assign domain specific name)

# Save the model as a '.lp' file.
model.write('model.lp')
```

Figure 7: Code skeleton for optimization model simulation.

```
**Problem Statement: {problem_name} in
{domain}**

**Background:**
(A brief description of background
information)

**Problem Description:**
(A brief description of {problem_name} in
{domain})

**Parameters:**
Only consider parameters listed below. And
these parameters will be provided in a
separated "data.json".
{parameter_skeleton}

**Decision Variables:**
(A list of decision variables and their
description)

**Objective:**
(State the objective function)

**Constraints:**
(A list of constraints in pure natural
language)

**Implementation Notes:**
(Any additional notes for implementation)

**Expected Outcome:**
(a brief description of the expected outcome)
```

Figure 8: Standard structure for word problem crafted from INFORMS AIMMS-MOPTA Optimization Modeling Competition.

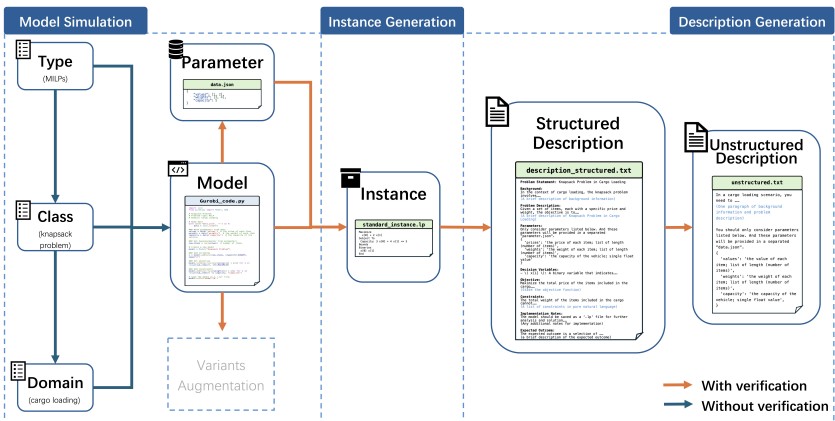

Figure 9: Data Construction Pipeline: Word problems are constructed through hierarchical reverse data evolution with model simulation, instance generation, and description generation. Processes with verification by either LLMs or human experts are marked in orange.

## C  RELATED WORK

**NLP for OR Modeling**    While substantial progress has been made in automatic modeling of general mathematical problems Bobrow (1964); Dellarosa (1986); Sundaram & Khemani (2015); Liu et al. (2025), work targeting operations research (OR) modeling has only recently begun to accelerate. Prior to the rise of LLMs, the NL4Opt competition Ramamonjison et al. (2022b) explored the feasibility of learning-based natural language interfaces for optimization solvers. With the advent of LLMs, several studies have demonstrated their potential as modeling assistants. Multi-agent and decomposition-based approaches—such as Holy Grail 2.0 Tsouros et al. (2023), Chain-of-Experts (CoE) Xiao et al. (2023), OptiMUS AhmadiTeshnizi et al. (2024), , OptiChat Chen et al. (2025), and GALA Cai et al. (2025). Similarly, focuses on bridging practitioner workflows with optimization models by enabling LLM-based interpretation, debugging, and reformulation. Complementary work such as NER4OPT Dakle et al. (2023) and Text2Zinc Singirikonda et al. (2025) investigates front-end components and datasets to structure natural language into entities, constraints, and MiniZinc specifications, thereby supporting more reliable text-to-model translation. Recent work on in-context constraint modeling Michailidis et al. (2024) further demonstrates that even general-purpose LLMs can produce executable constraint programs using retrieval and few-shot prompting alone. In parallel, Tang et al. (2024) showed that fine-tuning mid-size open-source models can yield substantial modeling improvements, underscoring the growing importance of domain-adapted OR modeling systems.

Beyond text-to-model translation, the OR community has also studied downstream solver-side automation techniques such as instance-specific algorithm configuration (ISAC) Kadioglu et al. (2010) and algorithm selection/scheduling Kadioglu et al. (2011), which leverage model-derived features to improve solver performance. Although these methods do not generate models from natural language, they underscore the value of structural information extracted from OR formulations—an insight that aligns with emerging LLM-based modeling pipelines.

Given the increasing use of LLMs for OR modeling, there is a strong need for benchmarks that reveal their capability boundaries Liu et al. (2024b); Zhou et al. (2024); Sawada et al. (2023). Several optimization modeling datasets have been proposed. The Linear Programming Word Problem (LPWP) dataset Ramamonjison et al. (2022a) spans multiple domains but focuses primarily on elementary-level LP problems. ComplexOR Xiao et al. (2023) introduces more sophisticated scenarios, yet its limited size and reliance on numeric data embedded within text still restrict the modeling complexity it captures. Other datasets—including IndustryOR Tang et al. (2024), MAMO Huang et al. (2024), and E-OPT Yang et al. (2024a)—use synthesis and augmentation to broaden coverage. The NLP4LP dataset AhmadiTeshnizi et al. (2024) attempts to separate numeric data from textual descriptions, but its problem scale remains small, and the descriptions are relatively structured with explicit variable-/constraint/objective declarations. In the constraint programming community, Text2Zinc Singirikonda et al. (2025) provides a large cross-domain dataset mapping natural-language descriptions to MiniZinc models, offering a valuable resource for studying general CP/OR text-to-model translation.

Compared to these benchmarks, our work aims to provide both a more comprehensive dataset and a more rigorous evaluation methodology, enabling a more precise assessment of LLM capabilities in optimization modeling.

**Modeling Equivalence Evaluation** The earliest work to evaluate NLP for OR modeling performance is to calculate the canonical accuracyRamamonjison et al. (2022a). This accuracy counts for the declaration-level(e.g., objective or constraints) matching score between predicted and reference formulations. This method has severe limitations as it's highly sensitive to superficial differences in formulation, such as variable naming or ordering. More recent benchmark works—including MAMOHuang et al. (2024), IndustryORTang et al. (2024), NLP4LPAhmadiTeshnizi et al. (2024), and OptiBenchYang et al. (2024b)—relies on solvers to assess modeling quality. They execute the predicted numerical models and compare the resulting optimal values with reference optimal values to evaluate correctness. While solver-based approaches better align with the functional goals of optimization, they introduce new limitations. The evaluation becomes dependent on solver behavior, which is often unstable, especially when the focus is on model structure equivalence rather than model instance outcome equivalence. For instance, small changes in parameters can render a model infeasible or non-convex, causing solvers to fail or return suboptimal solutions. As a result, optimal value mismatches may stem not from modeling errors but from solver or numerical issues, thereby confounding the reliability of equivalence assessment. Beyond optimal-value–based equivalence evaluation, Astorga et al. (2024) use satisfiability modulo theories (SMT) to assess formulation equivalence. SMT operates by encoding formulations into logical constraints and checking their satisfiability under the feasible region of the decision variables. However, such methods only work when the correspondence between decision variables is already provided, whereas identifying a correct mapping is often one of the most challenging parts of comparing two formulations. Zhai et al. (2025) attempt to prompt an LLM to infer a mapping between the decision variables of two formulations, followed by a solver-based verification step. If a valid mapping is found, the formulations are equivalent. While this offers a valuable perspective beyond objective-based evaluation, two limitations remain. First, LLM-generated mappings introduce computational overhead and may be unreliable, with hallucinated mappings or missing potential mappings. Second, the final verification still depends on a solver, which cannot reliably handle infeasible instances and may incur substantial computation when solvers struggle.

**Broader Research on AI for OR** Beyond model formulation, significant progress has been made in the field of AI for Operations Research (AI for OR), particularly in parameter generation and solving optimization problems Rajgopal (2004). In parameter generation, AI techniques have been employed for better simulation of key parameters of optimization problems Elmachtoub & Grigas (2022); Maragno et al. (2023); Bergman et al. (2022). Similarly, our work leverages LLMs to generate necessary problem data through a program of thoughts Chen et al. (2022a). On the optimization side, numerous studies have focused on leveraging AI models in automatic algorithm configuration Ansótegui et al. (2009); Lindauer et al. (2022); Anastacio & Hoos (2020), optimization algorithm selection Wang et al. (2019); Chi et al. (2022), and heuristic algorithm design Zeng et al. (2022); Talbi (2009); Romera-Paredes et al. (2024). Specifically, a line of research has modeled MILP/LP problems as bipartite graphs and applied Graph Neural Networks (GNNs) to make decisions at various stages of their solution processes Gasse et al. (2019); Zhou et al. (2020). These GNN-based methods have demonstrated efficacy in tasks such as variable selection and node branching, leading to significant improvements in solver performance. Inspired by this, we model optimization problems as bipartite graphs and formalize the evaluation paradigm based on the classical WL-test algorithm Leman & Weisfeiler (1968).

# D EQUIVALENCE EVALUATION

## D.1 MODEL EQUIVALENCE CLASS

**Definition D.1** (Model Equivalence). We say $\mathcal{C}(\mathcal{P})$ is a **model equivalence class** of the MILP/LP problem instance $\mathcal{P}$ if $\forall \hat{\mathcal{P}} \in \mathcal{C}(\mathcal{P}), \exists$ permutation matrices $P_1, P_2$ which shuffles the index of a vector or column index of a matrix s.t. $\hat{\mathcal{P}}$ can be written in the following form:

$$\min_x \hat{c}^T x,$$

$$\text{s.t. } \hat{A}x \hat{\circ} \hat{b}, \hat{l} \le x \le \hat{u}$$

where $\hat{b} = P_2 b, \hat{C} = P_1 C, \hat{A} = P_2 A P_1, \hat{\circ} = P_2 \circ, \hat{l} = P_1 l, u = P_1 u$.

$\forall \mathcal{P}_2 \in \mathcal{C}(\mathcal{P}_1)$, we say $\mathcal{P}_2$ is **model-equivalent** to $\mathcal{P}_1$, denote as $\mathcal{P}_1 \sim \mathcal{P}_2$.

## D.2 WEIGHTED BIPARTITE GRAPH FOR REPRESENTING MILP/LP

A weighted bipartite graph for a MILP/LP instance is denoted by $\mathbf{G} = (\mathbf{V} \cup \mathbf{W}, \mathbf{E})$, with vertex set $\mathbf{V} \cup \mathbf{W}$ divided into 2 groups $\mathbf{V} = \{\mathbf{v}_1, \cdots, \mathbf{v}_m\}$ for constraints, and $\mathbf{W} = \{\mathbf{w}_1, \cdots, \mathbf{w}_n\}$ for variables, $\mathbf{E}$ consisting of $E_{ij} = E(v_i, w_j), \forall i = 1, \cdots, m, j = 1, \cdots, n$. To fully represent all information in a MILP/LP instance, we associate each vertex with features:

- The constraint vertex $\mathbf{v}_i \in \mathbf{V}$ is equipped with a feature vector $\mathbf{H}^V$ with elements $\mathbf{h}_i^V = (b_i, o_i) \in \mathcal{H}^V = \mathbb{R} \times \{\le, \ge, =, <, >\}$
- The variable vertex $\mathbf{w}_j \in \mathbf{W}$ is equipped with a feature vector $\mathbf{H}^W$ with elements $\mathbf{h}_j^W = (c_j, \tau_j) \in \mathcal{H}^W = \mathbb{R} \times \{\mathbb{R} \cup -\infty\} \times \{\mathbb{R} \cup \infty\} \times \{0, 1\}$. $\tau_j = 1$ if $j \in \mathbb{Z}$ and $\tau_j = 0$ otherwise.

The edge $E_{ij} \in \mathbb{R}$ connects $\mathbf{v}_i \in \mathbf{V}$ and $\mathbf{w}_j \in \mathbf{W}$, $E_{ij} = \mathbf{A}_{ij}$. There is no edge connecting vertices in the same vertex group.

## D.3 CONNECTION BETWEEN MODEL EQUIVALENCE AND GRAPH ISOMORPHISM

To test whether 2 modeling instances were permutation equivalent, we can equivalently conduct isomorphism testing between their corresponding weighted bipartite graphs. Lemma D.1 establishes an equivalence between assessing modeling appropriateness and graph isomorphism testing.

**Definition D.2** (Graph Isomorphism). Consider 2 graphs $\mathcal{G}_1 = (\mathbf{G}_1, \mathbf{H}_1^V \times \mathbf{H}_1^W)$ and $\mathcal{G}_2 = (\mathbf{G}_2, \mathbf{H}_2^V \times \mathbf{H}_2^W)$ with $\mathbf{G}_i = (\mathbf{V}^i \cup \mathbf{W}^i, \mathbf{E}^i)|_{1 \le i \le 2}$. We say $\mathcal{G}_1$ and $\mathcal{G}_1$ are **isomorphic** if there exists permutation matrix $\mathbf{P}_1, \mathbf{P}_2$ such that: $\mathbf{P}_1 \mathbf{E}^1 P_2^T = \mathbf{E}^2, \mathbf{P}_1 \mathbf{H}_1^W = \mathbf{H}_2^W, \mathbf{P}_2 \mathbf{H}_1^V = \mathbf{H}_2^V$. If 2 graphs $\mathcal{G}_1$ and $\mathcal{G}_1$ are isomorphic, denote $\mathcal{G}_1 \overset{g}{\sim} \mathcal{G}_2$.

**Lemma D.1.** $\forall$ *MILP/LP instances* $\mathcal{P}_1, \mathcal{P}_2$ *with corresponding bipartite graph* $\mathcal{G}_1, \mathcal{G}_1$, *we have*

$$\mathcal{P}_1 \sim \mathcal{P}_2 \iff \mathcal{G}_1 \overset{g}{\sim} \mathcal{G}_2.$$

## D.4 PROOF OF LEMMA D.1:

We prove this lemma by proving 2 claims:

**Claim 1:** $\mathcal{G}_1 \sim \mathcal{G}_2 \implies \mathcal{P}_1 \sim \mathcal{P}_2$.

Suppose $\mathcal{G}_1 \sim \mathcal{G}_2$. For bipartite graphs $\mathcal{G}_1$ and $\mathcal{G}_2$, nodes $v_i$ would only connect with some node $w_j$ if the $j$-th constraint involves decision variable $x_i$. Therefore the adjacency matrix of $\mathcal{G}_k$ would be in the form $\mathbf{A}_{\mathbf{adj}}^{(\mathbf{k})} = \begin{bmatrix} 0 & \mathbf{A}_k^T \\ \mathbf{A}_k & 0 \end{bmatrix}, \forall k = 1, 2$. Now, by the assumption that $\mathcal{G}_1 \sim \mathcal{G}_2$, $\exists$ permutation matrix $\mathbf{P}$ such that

$$\mathbf{P} = \begin{bmatrix} \mathbf{P}_V & 0 \\ 0 & \mathbf{P}_W \end{bmatrix},$$

$$\mathbf{P}\mathbf{A}_{adj}^{(1)}\mathbf{P} = \mathbf{A}_{adj}^{(2)},$$

$$\mathbf{P}_V^T \mathbf{H}_1^V = \mathbf{P}^T \mathbf{H}_2^V,$$

$$\mathbf{P}_W^T \mathbf{H}_1^W = \mathbf{P}_W^T \mathbf{H}_2^W.$$

Therefore, we have

$$\mathbf{A}_{adj}^{(2)} = \begin{bmatrix} 0 & \mathbf{P}_V \mathbf{A}_1^T \\ \mathbf{P}_W \mathbf{A}_1 & 0 \end{bmatrix} \text{ and } \mathbf{H}_2 = \begin{bmatrix} \mathbf{P}_V \mathbf{H}_1^V \\ \mathbf{P}_W \mathbf{H}_1^W \end{bmatrix}.$$

We may reformulate the MILP/LP instance $\mathcal{P}_2$ as follows:

$$\mathcal{P}_2 : \quad \min_{\mathbf{x} \in \mathbb{R}^p \times \{0,1\}^{n-p}} \mathbf{c}^T \mathbf{P}_V \mathbf{x},$$
$$\text{s.t. } \mathbf{P}_W \mathbf{A} \mathbf{P}_V \mathbf{x} \circ \mathbf{P}_W \mathbf{b}, \mathbf{l} \leq \mathbf{P}_V \mathbf{x} \leq \mathbf{u},$$

By the definition of permutation equivalent, we say $\mathcal{P}_2 \sim \mathcal{P}_1$.

**Claim 2:** $\mathcal{P}_1 \sim \mathcal{P}_2 \implies \mathcal{G}_1 \sim \mathcal{G}_2$.

Suppose $\mathcal{P}_1 \sim \mathcal{P}_2$. By the definition of permutation equivalent class, $\exists$ permutation matrix $\mathbf{P}_1$ and $\mathbf{P}_2$ such that

$$\mathbf{A}_2 = \mathbf{P}_2 \mathbf{A}_1 \mathbf{P}_1$$
$$\mathbf{b}_2 = \mathbf{P}_2 \mathbf{b}_1,$$
$$\mathbf{C}_2 = \mathbf{P}_1^T \mathbf{C}_1,$$
$$\mathbf{P}_2 \circ_1 = \circ_2,$$

Therefore, the corresponding adjacent matrix in the bipartite graph of $\mathcal{P}_2$ is

$$\begin{aligned} \mathbf{A}_{adj}^{(2)} &= \begin{bmatrix} 0 & \mathbf{A}_2^T \\ \mathbf{A}_2 & 0 \end{bmatrix} \\ &= \begin{bmatrix} 0 & \mathbf{P}_1^T \mathbf{A}_1^T \mathbf{P}_2^T \\ \mathbf{P}_2 \mathbf{A}_1 \mathbf{P}_2 & 0 \end{bmatrix} \\ &= \begin{bmatrix} \mathbf{P}_1^T & 0 \\ 0 & \mathbf{P}_2 \end{bmatrix} \begin{bmatrix} 0 & \mathbf{A}_1^T \\ \mathbf{A}_1 & 0 \end{bmatrix} \begin{bmatrix} \mathbf{P}_1 & 0 \\ 0 & \mathbf{P}_2^T \end{bmatrix} \\ &= \hat{\mathbf{P}}^T \mathbf{A}_{adj}^{(1)} \hat{\mathbf{P}} \end{aligned}$$

In addition, we have $\mathbf{b}_2 = \mathbf{P}_2 \mathbf{b}_1, \mathbf{c}_2 = \mathbf{P}_1^T \mathbf{c}_1$. Therefore,

$$\mathbf{H}_2 = \begin{bmatrix} \mathbf{H}_2^V \\ \mathbf{H}_2^W \end{bmatrix} = \begin{bmatrix} \mathbf{P}_1^T & 0 \\ 0 & \mathbf{P}_2 \end{bmatrix} \begin{bmatrix} \mathbf{H}_1^V \\ \mathbf{H}_1^W \end{bmatrix} = \hat{\mathbf{P}}^T \mathbf{H}_1.$$

According to the definition of graph isomorphism, $\mathcal{G}_1$ is isomorphic to $\mathcal{G}_2$.

## D.5 Algorithms

---

**Algorithm 2** WL test for MILP/LP Graphs

---

**Require:** A graph instance $(G, H) \in \mathcal{G}_{m,n} \times \mathcal{H}_m^V \times \mathcal{H}_n^W$ and iterate limit $L > 0$.
1: Initialize with $C_i^{0,V} = HASH_{0,V}(h_i^V), C_j^{0,W} = HASH_{0,W}(h_j^W)$
2: **for** $l = 1, 2, \cdots, L$ **do**
3:     $C_i^{l,V} = HASH(C_i^{l-1,V}, \sum_{j=1}^n E_{i,j} HASH'_{l,W}(C_j)^{l-1,W})$
4:     $C_i^{l,W} = HASH(C_i^{l-1,W}, \sum_{j=1}^n E_{i,j} HASH'_{l,V}(C_j)^{l-1,V})$
5: **return** The multisets containing all colors $\{\{C_i^{L,V}\}\}_{i=0}^m, \{\{C_i^{L,W}\}\}_{j=0}^n$.

---

**Algorithm 3** Determine if the graph is symmetric decomposable

---

**Require:** Graph $\mathcal{G}$'s adjacent matrix $\mathbf{A}$ and type 2 stable partition sets of it's variable nodes $\mathcal{I} = \{I_1, I_2, \cdots, I_{s'}\}$ and constraint nodes $\mathcal{J} = \{J_1, J_2, \cdots, J_{t'}\}$.
**Ensure:** Returns **True** if the graph is decomposable symmetric; otherwise, **False**.
1: $k \leftarrow |I_1|$.
2: **if** $|I_s| \neq k$ or $|J_t| \neq k$ for some $s = 1, \cdots, s', t = 1, \cdots, t'$. **then**
3:     **return False**
4: **else**
5:     Initialize an empty Cluster dictionary $Cluster$
6:     **for** $i \leftarrow 0$ to $k - 1$ **do**
7:         $C \leftarrow$ the set of all numbers for type 2 stable partition sets
8:         Initialize an empty cluster set $Cluster[i]$, initialize an empty queue $Q$.
9:         **while** Set $C$ is not empty **do**
10:           **if** Q is empty **then**
11:             Randomly select a color $c \in C$, delete $c$ from $C$.
12:             $P_c \leftarrow$ the list of nodes labeled with $c \in C$.
13:             $Cluster[i] \leftarrow [P_c[i]]$, delete node $P_c[i]$ from $S$, push $P_c[i]$ in $Q$.
14:           **else**
15:             **while** Q not empty **do**
16:                $u \leftarrow Q.dequeue()$
17:                **for** neighborhood node $w$ of $u$ **do**
18:                   **if** w is not in any of $P_c$ or $w$ is in $Cluster[i]$ **then**
19:                     continue
20:                   **else if** $color(w)$ appears in $Cluster[i]$ **then**
21:                     **return False**
22:                   **else**
23:                     Add $w$ in $Cluster[i]$, delete $color(w)$ from $C$, push w in $Q$.
24:     **if** colors in $Cluster[i] \neq 1$ are not distinct for some $i = 0, \cdots k - 1$ **then**   $\triangleright$ check distinct color
25:         **return False**
26:     **else if** checkDisjointness($[S^1, \cdots, S^{k-1}]$) **then**         $\triangleright$ check disjointness
27:         **return False**
28:     **else if** checkConnectivity($[S^1, \cdots, S^{k-1}]$) **then**       $\triangleright$ check disconnectivity
29:         **return False**
30: **return** True

---

Notation: We denote the collection of all nodes $v_i's$ indexed by $i \in I_p$ as $\mathbf{I}_p$. Function checkDisjointness($[Cluster[1], \cdots, Cluster[k-1]]$) output True if any two sets $[i], Cluster[j], i \neq j$ shares common element, checkConnectivity($[Cluster[1], \cdots, Cluster[k-1]]$) output True if there exists some nodes $s \in Cluster[i], s' \in Cluster[j], i \neq j$ such that $s$ connected with $s'$.

## D.6 Proof Preparation for Theorem 3.1

Before establishing the proof, we first introduce the coloring refinement process of WL test for MILP/LP problem since it is the first step 1 in algorithm $\mathcal{A}$. For iteration $l$ of the algorithm we will be assigning to each node a tuple $H_i^L$ containing the node's old compressed label and a multiset of the node's neighbors' compressed labels. A multiset is a set (a collection of elements where order is not important) where elements may appear multiple times.

At each iteration $l$, we will additionally be assigning to each node a new "compressed" label $C_i^L$ with the same $H_i^L$ will get the same compressed label.

Repeat the above process for up to (m+n) (the number of nodes) iterations or until the partition of nodes by compressed label does not change from one iteration to the next, we will get a converged multiset.

In addition, we introduce preliminary tools for an algorithm-independent definition.

In fact, unfoldable and symmetric decomposable can be defined without relying on WL-test algorithm. We introduced equivalent definitions based on stable partition index sets.

**Definition D.3** (Stable Partition Index Sets). For a modeling instance $\mathcal{P}$ in the form of (1) with $n$ decision variables and $n$ constraints, define index set for optimization variables by $\mathcal{I} = \{I_1, I_2, \cdots, I_s\}$ and index set for constraints by $\mathcal{J} = \{J_1, J_2, \cdots, J_t\}$, where

- $\bigcup_{l=1}^s I_l = \{1, 2, \cdots, m\}$, $\bigcup_{k=1}^t J_k = \{1, 2, \cdots, n\}$;

- $I_{l_i} \cap I_{l_j} = \emptyset$, $J_{k_p} \cap J_{k_q} = \emptyset$, $\forall i, j \in [1, \cdots, |I_l|], i \neq j$, and $p, q \in [1, \cdots, |J_k|], p \neq q$.

We say $(\mathcal{I}, \mathcal{J})$ is a pair of stable partition index sets if the following condition holds:

1. $(c_i, \tau_i) = (c_{i'}, \tau_{i'}), \forall i, i' \in I_p$ for some $p \in 1, 2, \cdots, s$;

2. $(b_j, \circ_j) = (b_{j'}, \circ_{j'}), \forall j, j' \in J_q$ for some $q \in 1, 2, \cdots, t$;

3. $\forall p \in 1, 2, \cdots, s, q \in 1, 2, \cdots, t$, and $i, i' \in I_p$, we have $\sum_{j \in J_q} a_{ij} = \sum_{j \in J_q} a_{i'j}$;

4. $\forall p \in 1, 2, \cdots, s, q \in 1, 2, \cdots, t$, and $j, j' \in J_q$, we have $\sum_{i \in I_p} a_{ij} = \sum_{i \in I_p} a_{ij'}$;

**Lemma D.2.** *If there are no collision of hash functions and their weighted averages, then WL test algorithm 2 will finally terminated at some stable partition in $\mathcal{O}(m + n)$ iterations.*

Lemma D.2 is proved in Chen et al. (2022b).

**Definition D.4** (Unfoldable, by trivial partition). $\mathcal{P}$ is unfoldable if $\exists$ stable partition index sets $\mathcal{I}$ and $\mathcal{J}$ such that $\mathcal{I}$ or $\mathcal{J}$ are trivial partitions, i.e. $s = m$ and $t = n$.

**Definition D.5** (Decomposable Symmetric, by grouped partition). $\mathcal{P}$ is decomposable symmetric if the following condition holds:

$\exists$ stable partition index set $\mathcal{I}$ and $\mathcal{J}$ such that:

1. There are only two types of index set in $\mathcal{I}$ and $\mathcal{J}$. Type 1 set only contains a single index. Type 2 contains several indexes, denote type 2 sets by $I_1, \cdots, I_{s'}$; $J_1, \cdots, J_{t'}$. (By WL-test coloring, nodes with index in $I_i$ or $J_j$ share the same color.)

2. Type 2 sets $I_1, \cdots, I_{s'}$ and $J_1, \cdots, J_{t'}$ are equal-sized with $|I_p| = |J_q| = k > 1, \forall p \in \{1, 2, \cdots, s'\}$ and $q \in \{1, 2, \cdots, t'\}$.

3. There exist k disjoint groups $S^1, \cdots, S^k$ such that $|S^i \cap I_p| = |S^i \cap I_p| = 1$; and $\forall a \in S^i, b \in S^j$ with $i \neq j$, $a$ disconnected with $b$.

By Lemma D.2, we can show two sets of definitions are equivalent.

## D.7 Proof of Theorem 3.1

We construct the proof by two lemmas to illustrate sufficient conditions that the result of WL test coloring can reliably infer graph isomorphism.

**Lemma D.3.** *Suppose $\mathcal{P}_{standard}$ is unfoldable, then $\mathcal{G}_{standard}$ and $\mathcal{G}_{test}$ shares the same coloring $\Longleftrightarrow \mathcal{G}_{standard} \sim \mathcal{G}_{test}$.*

Suppose $\mathcal{P}_{standard}$ is unfoldable, want to show $\mathcal{A}(\mathcal{G}_{test}, \mathcal{G}_{standard}) ==$ Equivalent $\Longleftrightarrow \mathcal{P}_{test} \sim \mathcal{P}_{standard}$.

If $\mathcal{P}_{test} \sim \mathcal{P}_{standard}$, it is trivial that $\mathcal{A}(\mathcal{G}_{test}, \mathcal{G}_{standard}) ==$ Equivalent.

Now, consider when $\mathcal{A}(\mathcal{G}_{test}, \mathcal{G}_{standard}) ==$ Equivalent and $\mathcal{P}_{standard}$ unfoldable, we have $len(\mathbb{A}_1) = len(\mathcal{C}_1)$ & $len(\mathbb{A}_2) = len(\mathcal{C}_2)$.

By the detection algorithm, every color in the multisets output by WL test must be distinct, and multisets for $\mathcal{P}_{standard}$ are the same as multisets for $\mathcal{P}_{standard}$. One stable partition of $\mathcal{G}_{standard}$ and is $\{I_1, \cdots, I_n\}, \{J_1, \cdots, J_m\}$, where $I_k, J_l$ are single-element sets. WLOG, assume $I_k = i_k, J_l = j_l$.

Similarly, denote the stable partition of $\mathcal{G}_{test}$ by $\{I'_1, \cdots, I'_n\}, \{J'_1, \cdots, J'_m\}$, with $I'_k = [i'_k], J'_l = [j'_l]$.

Now, define a bijection mapping that shuffles $[i_1, \cdots, i_m]$ and $[j_1, \cdots, j_n]$ to get $[i'_1, \cdots, i'_m]$ and $[j'_1, \cdots, j'_n]$, denote such mapping by $\mathbf{P}$. (Since each element in $[i_1, \cdots, i_m], [j_1, \cdots, j_n], [i'_1, \cdots, i'_m]$, or $[j'_1, \cdots, j'_n]$ is distinct, we can uniquely find such bijection).

Notice that such bijection may only map the index of $v_i^{standard}$ to the index of $v_j^{test}$ and map the index of $w_l^{standard}$ to the index of $w_p^{test}$, we can separately define a bijection for decision variable index as $\mathbf{P}_1$ and a bijection for constraint index as $\mathbf{P}_2$.

Therefore, exists bijection $\mathbf{P}_1$ and $\mathbf{P}_2$ such that $\mathcal{P}_{test}$ can be written in the following form:

$$\min_x \hat{c}^T x,$$

$$\text{s.t. } \hat{A}x \hat{\diamond} \hat{b}$$

where $\hat{b} = P_2 b_{standard}, \hat{C} = P_1 C_{standard}, \hat{A} = P_2 A_{standard} P_1, \hat{\diamond} = P_2 \diamond_{standard}$. This implies $\mathcal{P}_{test} \sim \mathcal{P}_{standard}$.

**Lemma D.4.** *Suppose $\mathcal{P}_{standard}, \mathcal{P}_{test}$ are decomposible symmetric, then $\mathcal{G}_{standard}$ and $\mathcal{G}_{test}$ shares the same coloring $\Longleftrightarrow \mathcal{G}_{standard} \sim \mathcal{G}_{test}$.*

When $\mathcal{P}_{standard}$ is decomposible symmetric, and algorithm $\mathcal{A}$ output "Equivalent", the partition sets of $\mathcal{G}_{standard}$ and $\mathcal{G}_{test}$ can be denoted as

$$\mathcal{I}_{standard} = [I_1, \cdots, I_k, I_{k+1}, \cdots, I_s];$$
$$\mathcal{J}_{standard} = [J_1, \cdots, J_l, J_{l+1}, \cdots, J_t];$$
$$\mathcal{I}_{test} = [\hat{I}_1, \cdots, \hat{I}_k, \hat{I}_{k+1}, \cdots, \hat{I}_s];$$
$$\mathcal{J}_{test} = [\hat{J}_1, \cdots, \hat{J}_l, \hat{J}_{l+1}, \cdots, \hat{J}_t],$$

where set $[I_1, \cdots, I_k], [\hat{I}_1, \cdots, \hat{I}_k], [J_1, \cdots, J_l], [\hat{J}_1, \cdots, \hat{J}_l]$, only contains one index, and set $[I_{k+1}, \cdots, I_s], [\hat{I}_{k+1}, \cdots, \hat{I}_s], [J_{k+1}, \cdots, J_t], [\hat{J}_{k+1}, \cdots, \hat{J}_t]$ consist at least 2 indexes; $I_i, \hat{I}_i$ shares the same color $\forall i$; and $J_j, \hat{J}_j$ shares the same color $\forall j$.

Now, define a bijection mapping that maps $[I_1, \cdots, I_k, I_{k+1}, \cdots, I_s, J_1, \cdots, J_l, J_{l+1}, \cdots, J_t]$ to $[\hat{I}_1, \cdots, \hat{I}_k, \hat{I}_{k+1}, \cdots, \hat{I}_s, \hat{J}_1, \cdots, \hat{J}_l, \hat{J}_{l+1}, \cdots, \hat{J}_t]$, by the following rules:

1. For the unique index $i \in I_p$, where $p \in \{1, \cdots, k\}$, map $i$ to the unique index $i' \in \hat{I}_p$.

2. For the unique index $j \in J_q$, where $q \in \{1, \cdots, l\}$, map $j$ to to the unique index $j' \in \hat{J}_q$.

3. For the remaining nodes, we consider a cluster-wise mapping, i.e finding some equivalent clusters, mapping a cluster to another, and providing a unique mapping rule within choosen cluster.

Let $V'$ and $\hat{V}'$ be sets of all variable nodes except those with unique color in $\mathcal{G}_{standard}$ and $\mathcal{G}_{test}$; $W'$ and $\hat{W}'$ be sets of all constraint nodes except those with unique color in $\mathcal{G}_{standard}$ and $\mathcal{G}_{test}$. Find clusters $S^1, \cdots, S^r$ such that each $S^i$ is disconnected, disjoint, consists of nodes with the same combination of unique colors as other $S^{i'}s$, and $\bigcup_{i=1}^r S^i = V' \cup W'$. Similarly, for symmetric decomposable $\mathcal{G}_test$ with the same coloring distribution, we can find clusters $\hat{S}^1, \cdots, \hat{S}^r$ such that $\hat{S}^i$ has the same coloring distribution with $S^i$, and each $S^i$ is disconnected, disjoint, consists of nodes with the same combination of unique colors as other $S^{i'}s$, and $\bigcup_{i=1}^r \hat{S}^i = \hat{V}' \cup \hat{W}'$.

The existence of $S^1, \cdots, S^r$ and $\hat{S}^1, \cdots, \hat{S}^r$ are guaranteed by the symmetric decomposable property of $\mathcal{G}_{standard}$ and $\mathcal{G}_{test}$.

Now, we can define a bijection that maps $S^i$ to a corresponding cluster $\hat{S}_i$. Note that nodes in one cluster have distinct colors. The bijection mapping maps nodes from cluster $S^i$ to $\hat{S}_i$ according to color-matching, i.e. a node maps to another one when they are in the same color.

Now, consider the adjacency matrix of the representing bipartite graph

$$\mathbf{A}_{adj} = \begin{bmatrix} 0 & \mathbf{A}^T \\ \mathbf{A} & 0 \end{bmatrix}.$$

Since node groups $S^1, \cdots, S^k$ are disconnected, we can rearrange matrix $A$ by some column permutation $\mathbf{P}_1^b$ and row permutation $\mathbf{P}_2^b$ such that

$$\mathbf{P}_1^b \mathbf{A} \mathbf{P}_2^b = \begin{bmatrix} \mathbf{A}_1 & 0 & \cdots & 0 & \mathbf{a_1} \\ 0 & \mathbf{A}_2 & \cdots & 0 & \mathbf{a_2} \\ \vdots & \vdots & \ddots & \vdots & \vdots \\ 0 & 0 & \cdots & \mathbf{A}_r & \mathbf{a_r} \\ \mathbf{b_1}^T & \mathbf{b_2}^T & \cdots & \mathbf{b_r}^T & \mathbf{A}_{r+1} \end{bmatrix}$$

where $A_1, \cdots, A_r$ are coefficient matrix for r clusters $S^1, \cdots, S^r$ with associated decision variables and constraints, and $A_{r+1}$ is a $k \times l$ matrix.

The above composition of bijection mapping operations is equivalent to applying permutation operations on $A_{standard}, b_{standard}, c_{standard}, \circ_{standard}$ by the following steps:

1. point-wise mapping for variables: permute $c$ and $A$ by permutation matrix $\mathbf{P}_1^0$ to map unique index $i \in I_p$ to $i' \in \hat{I}_p$, which produce $\hat{c} = \mathbf{P}_1^0 c_{standard}$ and $\hat{A} = A_{standard} \mathbf{P}_1^0$

2. point-wise mapping for constraints: permute $b_{standard}, \circ_{standard}$ and $\hat{A}$ by permutation matrix $\mathbf{P}_2^0$ to map unique index $j \in I_q$ to $j' \in \hat{J}_q$, which produce $\hat{b} = \mathbf{P}_2^0 b_{standard}$, $\hat{\circ} = \mathbf{P}_2^0 \circ_{standard}$ and $\hat{A} = \mathbf{P}_2^0 \hat{A} = \mathbf{P}_2^0 A_{standard} \mathbf{P}_1^0$

3. clustering mapping: permute $\hat{c}$ and $\hat{A}$ by permutation matrix $\mathbf{P}_1^c$ and permute $\hat{b}, \hat{\circ}$ and $\hat{A}$ by permutation matrix $\mathbf{P}_2^c$ to produce

$$\hat{A} = \mathbf{P}_1^c \hat{A} \mathbf{P}_2^c = \begin{bmatrix} \mathbf{A}_1 & 0 & \cdots & 0 & \mathbf{a_1} \\ 0 & \mathbf{A}_2 & \cdots & 0 & \mathbf{a_2} \\ \vdots & \vdots & \ddots & \vdots & \vdots \\ 0 & 0 & \cdots & \mathbf{A}_r & \mathbf{a_r} \\ \mathbf{b_1}^T & \mathbf{b_2}^T & \cdots & \mathbf{b_r}^T & \mathbf{A}_{r+1} \end{bmatrix}$$

$$= \mathbf{P}_1^c \mathbf{P}_1^0 A_{standard} \mathbf{P}_2^0 \mathbf{P}_2^c,$$

$\hat{b} = \mathbf{P}_2^c \mathbf{P}_2^0 b_{standard}$, $\hat{\circ} = \mathbf{P}_2^c \mathbf{P}_2^0 \circ_{standard}$, and $\hat{c} = \mathbf{P}_1^c \mathbf{P}_1^0 c_{standard}$

4. in-cluster mapping: iteratively permute $\hat{c}$ and $\hat{A}$ by permutation matrices $\mathbf{P}_1^1, \cdots, \mathbf{P}_1^r$ and permute $\hat{b}, \hat{\circ}$ and $\hat{A}$ by permutation matrices $\mathbf{P}_2^1, \cdots, \mathbf{P}_2^r$ to produce $\hat{A} = \mathbf{P}_1^r \cdots \mathbf{P}_1^1 \mathbf{P}_1^c \mathbf{P}_1^0 A_{standard} \mathbf{P}_2^0 \mathbf{P}_2^c \mathbf{P}_2^1 \cdots \mathbf{P}_2^r$, $\hat{b} = \mathbf{P}_2^k \cdots \mathbf{P}_2^1 \mathbf{P}_2^b \mathbf{P}_2^0 b_{standard}$, $\hat{\circ} = \mathbf{P}_2^r \cdots \mathbf{P}_2^1 \mathbf{P}_2^b \mathbf{P}_2^0 \circ_{standard}$, and $\hat{c} = \mathbf{P}_1^r \cdots \mathbf{P}_1^1 \mathbf{P}_1^c \mathbf{P}_1^0 c_{standard}$

Now, define $\mathbf{P}_1 = \mathbf{P}_1^k \cdots \mathbf{P}_1^1 \mathbf{P}_1^c \mathbf{P}_1^0$ and $\mathbf{P}_2 = \mathbf{P}_2^k \cdots \mathbf{P}_2^1 \mathbf{P}_2^c \mathbf{P}_2^0$, we can write $\mathcal{P}_{test}$ in the following form:

$$\min_x \hat{c}^T x,$$

$$\text{s.t. } \hat{A}x \hat{\circ} \hat{b}$$

where $\hat{b} = P_2 b_{standard}, \hat{C} = P_1 C_{standard}, \hat{A} = P_2 A_{standard} P_1, \hat{\circ} = P_2 \circ_{standard}$. This implies $\mathcal{P}_{test} \sim \mathcal{P}_{standard}$.

## D.8 COMPLEXITY ANALYSIS

For the two main types of problem realizations in our benchmark, Algorithm 2 converges in $\mathcal{O}(m+n)$ iteration. In addition, for problems with $m$ variables and $n$ constraints, the time complexity to distinguish tested problem realizations from the standard realization is at most $\mathcal{O}(kmn)$, which is is significantly lower than classical algorithms employed by popular solvers, such as simplex method for LP and branch and bound algorithm for MILP. Specifically,

1. **For unfoldable problem instances**, algorithm 2 converges in at most $\mathcal{O}(m+n)$ iterations according to lemma D.2.

2. **For decomposable symmetric problem instances**, algorithm 2 converges in at most $\mathcal{O}(m+n)$ iterations, and we shall further conduct symmetric decomposable detection using algorithm 3, which takes time complexity $\mathcal{O}(kmn)$ in the worst case, where $k$ is the number of clusters in the symmetric decomposable graph. The total time complexity could be $\mathcal{O}(kmn)$.

## D.9 RANDOMLY SAMPLING SUFFICES TO OBTAIN SYMMETRIC DECOMPOSABLE

To make WL test work, it is desirable to sample a symmetric decomposable instance. In Theorem D.5 and D.6, we proved that for a large range of modeling problems with reasonable assumptions, we can sample a symmetric decomposable instance from its **parameter support** with probability 1.

**Definition D.6** (Modeling Parameter Support). For a class of model formulation $\mathcal{M}$ with $n$ decision variables and $m$ constraints, the **parameter set** $\Theta(\mathcal{M})$ is a collection of all possible values for instance parameter $(\mathbf{A}, \mathbf{c}, \mathbf{b}, \circ, \tau)$. The parameter set associated with decision variable $x_i$ is $\Theta(\mathcal{M}, i) = \left\{ [\mathbf{A}_{:,i}^T, c_i] \right\}$.

An example of a formulation parameter support is attached in Appendix Section E.

**Theorem D.5** (Efficient Sampling - continuous case). *Suppose a model $\mathcal{M}$ satisfies the following conditions:*

*For each $\vec{\theta}_i \in \mathbb{R}^d, i = 1, \cdots, n$, there exists a coordinate $k_i$ such that $\vec{e}_{k_i}^\top \vec{\theta}_i$ follows a continuous distribution $\mu_i$ independently across $i$.*

*then a random draw $\vec{\theta} \sim \Theta$ yields a **symmetric decomposable** instance $\mathcal{M}(\vec{\theta})$ almost surely.*

**Theorem D.6** (Efficient Sampling - discrete case). *Suppose a model $\mathcal{M}$ satisfies the following conditions:*

*$\forall i = 1, \cdots, n, \forall \vec{\theta}_i \in \mathbf{R}^d, \exists k_i \in \{1, \cdots, d\}$ such that $\vec{e}_{k_i}^\top \vec{\theta}_i \sim \mu_i(\cdot)$ and independent of the distribution of $\vec{\theta}_j, \forall j \neq i$, where $\vec{e}_{k_i}$ is the $k_i$-th standard basis vector in $\mathbf{R}^d, \mu_i(\cdot)$ is some discrete uniform distribution with $u_i(\vec{e}_{k_i}^\top \vec{\theta}_i) \sim Uniform\{x_1 \cdots x_l\}$, i.e. at lease one coordinate of $\vec{\theta}_i$ can be randomly sampled with probability $\frac{1}{l}$, where $k_i$ is the index of coordinate in $\vec{\theta}_i$ that being sampled.*

*Then, as $l \to \infty$, randomly sample $\vec{\theta}$ from parameter support $\Theta$, we can get a symmetric decomposable instance for model $\mathcal{M}$ with probability 1.*

We present the proof for Theorem D.5 and D.6 in Appendix D.10.

### D.10  PROOF OF THEOREM D.5 AND THEOREM D.6

**Proof:**

**Lemma D.7.** *Suppose model $\mathcal{M}$ satisfies the following assuption:*

*$\forall i = 1, \cdots, n. \forall \vec{\theta}_i \in \mathcal{R}^d, \exists k_i \in \{1, \cdots, d\}$ such that $\vec{e}_{k_i}^{\top} \vec{\theta}_i \sim \mu_i\left(\vec{\theta}_i\right)$ and independent of the distribution of $\vec{\theta}_j$, where $\vec{e}_{k_i}$ is the $k_i$-th standard basis vertor in $\mathbf{R}^d$, $\mu_i(\vec{\theta}_i)$ is some continuous distribution; i.e., at least one coordinate of $\vec{\theta}_i$ can be randomly sampled according to some continuous distribution.*

*Then, we have $P(\vec{\theta}_i = \vec{\theta}_j) = 0, \forall i \neq j.$*

Proof of lemma D.7:

Consider $i \neq j, 0 \leq P\left(\vec{\theta}_i = \vec{\theta}_j\right) \leq P\left(\vec{e}_{k_j}\vec{\theta}_j = \vec{e}_{k_j}\vec{\theta}_i\right) = 0$ since $\mu_i$ is continuous distribution.

By lemma D.7, we have $P\left(\vec{\theta}_i \neq \vec{\theta}_j\right) = 1.$

**Lemma D.8.** *Suppose model $\mathcal{M}$ satisfies the following condition:*

*$\forall i = 1, \cdots, n, \forall \vec{\theta}_i \in \mathbf{R}^d, \exists k_i \in \{1, \cdots, d\}$ such that $\vec{e}_{k_i}^{\top}\vec{\theta}_i \sim \mu_i(\cdot)$ and independent of the distribution of $\vec{\theta}_j, \forall j \neq i$, where $\vec{e}_{k_i}$ is the $k_i$-th standard basis vertor in $\mathbf{R}^d$, $\mu_i(\cdot)$ is some discrete uniform distribution with $u_i(\vec{e}_{k_i}^{\top}\vec{\theta}_i) \sim Uniform\{x_1 \cdots x_l\}$, i.e. at lease one coordinate of $\vec{\theta}_i$ can be randomly sampled with probability $\frac{1}{l}$, where $k_i$ is the index of coordinate in $\vec{\theta}_i$ that being sampled.*

*Then $P(\vec{\theta}_i = \vec{\theta}_i) \to 0$ as $l \to \infty$.*

Proof of lemma D.8:

$$
\begin{aligned}
P\left(\vec{\theta}_i = \vec{\theta}_j\right) &= P\left(\vec{e}_{k_i}\vec{\theta}_i = \vec{e}_{k_i}\vec{\theta}_j\right) \\
&= \sum_x P\left(\vec{e}_{k_i}\vec{\theta}_j = x \mid \vec{e}_{k_i}\vec{\theta}_i = x\right) P\left(\vec{e}_{k_i}\vec{\theta}_i = x\right) \\
&= \sum_x P\left(\vec{e}_{k_i}\vec{\theta}_j = x\right) \\
&= \sum_x \frac{1}{l^2} \\
&= \frac{1}{l}.
\end{aligned}
$$

as $l \to \infty, \quad P\left(\vec{\theta}_i = \vec{\theta}_j\right) \to 0.$

**Lemma D.9.** *Suppose a modeling instance $\mathcal{P}$ has $P(\vec{\theta}_j = \vec{\theta}_{j'}) = 0, \forall j \neq j'$, then $P(\mathcal{P}$ is symmetric decomposable$) = 1.$*

$P(\vec{\theta}_j \neq \vec{\theta}_{j'}) = 1,$ for all $j \neq j'$

Proof of lemma D.9:

Suppose $\exists$ index set $k \subset \{1, 2, \cdots, d\}$ such that $\forall k \in k, \vec{e}_k\vec{\theta}_j \neq \vec{e}_k\vec{\theta}_j$, want to show the joint probability of the following event is 1 :

1. Event A: $c_j \neq c_{j'}$. [Objective coefficients are not the same.]
2. Event B: $\sum_{i \in I} a_{ij} \neq \sum_{i \in I} a_{ij'}$ [accumulated edge weights for variable nodes of $j, j'$ are not the same].
3. Event C: $\sum_{q \in J} a_{i'q} \neq \sum_{q \in J} a_{iq}$ for some $J$ containing index $j$ or $j'$ and some $i \neq i' \in I$. [accumulated edge weights for two constraint nodes are not the same];

where $I$ and $J$ are sets in stable partitions $\mathcal{I}, \mathcal{J}$. It is euqivalent to show $P(A \cup B \cup C) = 1$. Now, consider two cases when $j, j' \in \{1, \cdots, n\}$:

1. Case 1: $\exists k \in K$ s.t. $\vec{e}_k^\top \vec{\theta}_j = c_j, \vec{e}_k^\top \vec{\theta}_{j'} = c_{j'}$, then $c_j \neq c_{j'}$.

2. Case 2: $\exists k \in K$ s.t. $\vec{e}_k^\top \vec{\theta}_j = a_{ij}, \vec{e}_k^\top \vec{\theta}_{j'} = a_{ij'}$ for some $i$, then $a_{ij} \neq a_{ij'}$ for some $i$.

Notice that $P(\Omega) = P(\text{Case } 1 \cup \text{Case } 2) = 1)$.

It suffices to show $P(A \cup B \cup C \mid \text{case } 1 \cup \text{case } 2) = 1$. Now, $P(A \cup B \cup C \mid \text{case } 1) = 1$ since $P(A \mid \text{case } 1) = 1$. It suffices to show $P(A \cup B \cup C \mid \text{case } 2) = 1$; Is suffices to show $P(B \cup C \mid \text{case } 2) = 1$.

Now, suppose $\exists R \in K$ s.t. $\vec{e}_k \vec{\theta}_j = a_{ij} \neq \vec{e}_k \vec{\theta}_{j'} = a_{ij'}$.

Consider $I$ containing $i$. WLOG, suppose $\hat{I} \subset I$ is an index set that containing all $i's$ such that $a_{ij} \neq a_{ij'}$, and $\sum_{i \in I/\hat{I}} (a_{ij} - a_{ij'}) = c$ for come constant $c$, then

$$P\left(\sum_{i \in I} a_{ij} \neq \sum_{i \in I} a_{ij'}\right) = P\left(I_{i \in \hat{I}} a_{ij} \neq \sum_{i \in I} a_{ij'} + c\right)$$

$$= 1 - P\left(I_{i \in \hat{I}} a_{ij} = \sum_{i \in I} a_{ij'} + c\right)$$

$$= 1 - 0$$

$$= 1$$

The third equality holds since $\sum_{i \in \hat{I}} a_{ij}$ and $\sum_{i \in \hat{I}} a_{ij'}$ are independent and can be sampled from some continuous distribution. Therefore. $P(B \mid \text{case } 2) = 0$, we have $P(A \cup B \cup C) = P(A \cup B \cup C \mid \Omega) = P(A \cup B \cup C \mid (\text{case } 1 \cup case2)) = 1$.

$$P(A \cup B \cup C) = P(A \cup B \cup C \mid \Omega) = P(A \cup B \cup C \mid \text{ case } 1 \cup \text{ case } 2) = 1.$$

Now, by lemma D.7 and lemma D.9, we can prove Theorem D.5; by lemma D.8 and lemma D.9, we can prove Theorem D.6.

# E EXAMPLES

## E.1 EXAMPLES FOR LIMITATIONS OF SOLVER-BASED EVALUATION

**Example 1** (The solver returns values, and the execution accuracy is 1 but the mathematical model is actually wrong). *Consider a car production and revenue maximization problem. A manufacturer produces two types of cars: sedans and SUVs. Let the decision variables of $x$ be the number of sedans to produce and $y$ be the number of SUVs to produce. The correct formulation is:*

*Maximize $30x + 50y$*
*such that: $x + 2y \leq 100$ (Production capacity in labor-hours)*
*$x \geq 0, y \geq 0$ (Non-negativity)*

*Now suppose an LLM generates an incorrect model with an additional erroneous constraint:*

*Maximize $30x + 50y$*
*such that: $x + 2y \leq 100$ (Production capacity in labor-hours)*
*$x + y \leq 40$ (ERRONEOUS constraint)*
*$x \geq 0, y \geq 0$ (Non-negativity)*

*If we test with a data configuration $\theta$ where production capacity = 80 and the market demand limit is 40, both models will yield the same optimal solution and optimal value: produce 40 SUVs for a*

*revenue of $2,000. However, if the data configuration changes to $\theta'$ with production capacity = 200, the correct model would recommend producing 100 SUVs for a revenue of $5,000, while the incorrect model would still limit production to 40 units total due to the erroneous constraint.*

**Example 2** (The solver returns constant value, and its useless for modeling equivalence detection)**.** *Consider a facility location problem where the goal is to determine whether it is possible to open a subset of facilities to serve all customer demand within a fixed budget. The objective is a constant (e.g., 0), since only feasibility is of interest:*

$$\text{Minimize } 0$$

$$\text{such that:} \quad \sum_{j \in \mathcal{F}} x_j \cdot c_j \leq B \quad \text{(Budget constraint)}$$

$$\sum_{j \in \mathcal{F}} a_{ij} x_j \geq 1 \quad \forall i \in \mathcal{D} \quad \text{(Coverage: each demand point must be served)}$$

$$x_j \in \{0, 1\} \quad \forall j \in \mathcal{F}$$

*Now, suppose the LLM generates an incorrect model with slightly relaxed constraints:*

$$\text{Minimize } 0$$

$$\text{such that:} \quad \sum_{j \in \mathcal{F}} x_j \cdot c_j \leq B \quad \text{(Same budget constraint)}$$

$$\sum_{j \in \mathcal{F}} a_{ij} x_j \geq 0.5 \quad \forall i \in \mathcal{D} \quad \text{(ERRONEOUS weaker coverage)}$$

$$x_j \in \{0, 1\} \quad \forall j \in \mathcal{F}$$

*If both models happen to be feasible under a specific data configuration $\theta$ (e.g., a small number of facilities with low costs and high coverage), then the solver will return "feasible" for both. However, the second model allows partial coverage (due to the threshold of 0.5), which violates the intended semantics. Since the objective function is constant, execution accuracy based on solver output cannot detect this structural mistake.*

**Example 3** (The mathematical model is incorrect but the execution accuracy is invalid for infeasible problems)**.** *Consider the same car production problem, but with modified constraints:*

$$\text{Maximize } 30x + 50y$$

$$\text{such that: } x + 2y \leq 10 \quad \text{(Limited production capacity in labor-hours)}$$

$$x \geq 20, y \geq 0 \quad \text{(Minimum sedan production requirement)}$$

*This correct model is genuinely infeasible because the minimum sedan production requirement ($x \geq 20$) cannot be satisfied with the limited production capacity ($x + 2y \leq 10$). Now suppose an LLM generates an incorrect model with an erroneous constraint:*

$$\text{Maximize } 30x + 50y$$

$$\text{such that: } x + 2y \leq 10 \quad \text{(Limited production capacity in labor-hours)}$$

$$x \geq 5, y \geq 7 \quad \text{(ERRONEOUS minimum requirements)}$$

$$x, y \geq 0 \quad \text{(Non-negativity)}$$

*For the data configuration $\theta$ shown above, both models will be evaluated as infeasible by the solver. The execution accuracy metric cannot distinguish between the correct model that is genuinely infeasible under this configuration and the incorrect model that is infeasible due to contradictory constraints ($x \geq 5$ and $y \geq 7$ would require at least 19 labor-hours, exceeding the 10 available).*

E.2 EXAMPLES FOR MODEL, MODEL PARAMETER SET, AND MODEL INSTANCE

**Example 4** (Problem Data Support)**.** *Consider the following problem: A farmer who seeks to maximize profit using $t$ acres of land to grow wheat and rice. Wheat yields $m_1$ kilograms per acre and*

*rice yields $m_2$ kilograms per acre; each kilogram of wheat brings a profit of $c_1$, and each kilogram of rice brings a profit of $c_2$. Then the problem data support is:*

$$\Theta = T \times M_1 \times M_2 \times C_1 \times C_2 \subseteq \mathbb{R}_+^5,$$

*containing all possible problem data $(t, m_1, m_2, c_1, c_2)$.*

*A ground-truth model is:*

$$\mathcal{M} : (t, m_1, m_2, c_1, c_2) \mapsto \quad \begin{array}{ll} \min_{x_1, x_2} & -m_1 c_1 x_1 - m_2 x_2 c_2 \\ s.t. & x_1 + x_2 \leq t, \\ & x_1 \geq 0, \\ & x_2 \geq 0. \end{array}$$

**Example 5** (Model Parameter Set for Blending Problem). *For example, a blending problem can be formulated as:*

$$\min_x \sum_{i=1}^n c_i x_i$$

$$s.t. \sum_{i=1}^n a_{ji} x_i \geq p_j, \forall j = 1, \cdots, m.$$

$$x_i \leq u_i, \forall i = 1, \cdots, n.$$

*The corresponding parameter set $\Theta(\mathcal{M}_{blend})$ can be defined as*

$$\Theta(\mathcal{M}_{blend}) = \Big\{ (\mathbf{A}, \mathbf{c}, \mathbf{b}, \circ) \Big| \mathbf{A} = [\hat{\mathbf{A}}^T, I_n]^T, \text{ where } \hat{\mathbf{A}} \in \mathbb{R}^{m \times n} \text{ and } I_n \text{ is an } n \times n$$

$$\text{identity matrix}; \mathbf{c} = [c_1, \cdots, c_n]^T \in \mathbb{R}^n; \mathbf{b} = [-p_1, \cdots, -p_J, -u_1, \cdots, -u_n]^n \in \mathbb{R}^{m+n};$$

$$\circ = [\geq, \cdots, \geq, \leq, \cdots, \cdots, \leq]_{1 \times (m+n)}^T \Big\}.$$

*The parameter set associated with $x_i$ is $\Theta(\mathcal{M}_{blend}, i) = \big\{ [\mathbf{A}_{:,i}^T, c_i] \big\} = \mathbb{R}^{m+1}$.*

### E.3 EXAMPLES FOR SYMMETRY

**Example 6** (Undesirable Symmetry). *Discriminating problem instances involving symmetry in their decision variables or constraints can be tricky. Because some non-isomorphic bipartite graphs cannot be distinguished by WL-test due to their automorphic structure in the graph. For example, Chen et al. (2022b) illustrates one case in which two MILP graphs are non-isomorphic while WL-test outputs the same multiset.*

$$\min_{x \in \mathbb{R}^6} x_1 + x_2 + x_3 + x_4 + x_5 + x_6,$$

$$\text{s.t. } x_1 + x_2 = 1, \ x_2 + x_3 = 1, \ x_3 + x_4 = 1,$$

$$x_4 + x_5 = 1, \ x_5 + x_6 = 1, \ x_6 + x_1 = 1,$$

$$0 \leq x_j \leq 1, \ x_j \in \mathbb{Z}, \ \forall j \in \{1, 2, \dots, 6\}.$$

$$\min_{x \in \mathbb{R}^6} x_1 + x_2 + x_3 + x_4 + x_5 + x_6,$$

$$\text{s.t. } x_1 + x_2 = 1, \ x_2 + x_3 = 1, \ x_3 + x_1 = 1,$$

$$x_4 + x_5 = 1, \ x_5 + x_6 = 1, \ x_6 + x_4 = 1,$$

$$0 \leq x_j \leq 1, \ x_j \in \mathbb{Z}, \ \forall j \in \{1, 2, \dots, 6\}.$$

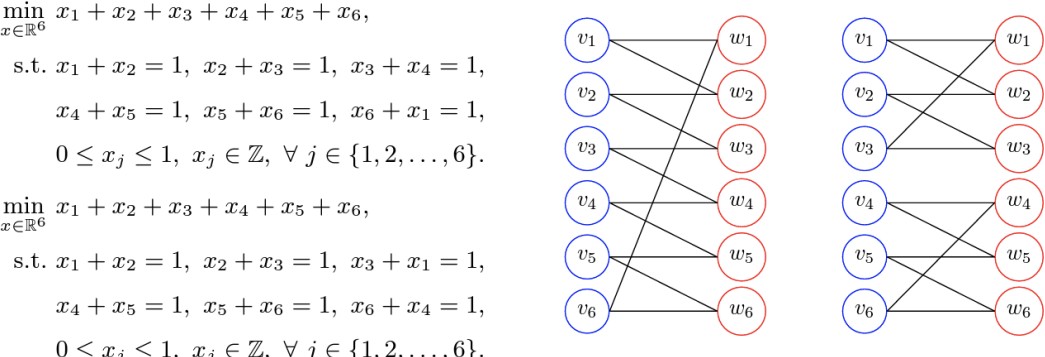

Figure 10: Two non-isomorphic MILP graphs that cannot be distinguished by the WL test

**Example 7** (Symmetric Decomposable Problem: Bin-Packing). *For decomposable symmetric problems, their corresponding bipartite graph can be divided into several symmetric sub-graphs, each*

*isomorphic and disconnected from the others. For example, an instance of bin-packing with heterogeneous vehicles is formulated as*

$$\min_{x\in\{0,1\}^q, y\in\{0,1\}^p} \sum_{j=1}^{p} y_j$$

$$s.t. \sum_{i} s_i x_{ij} \leq b y_j, \forall j = 1, \cdots, p.$$

$$\sum_{j=1}^{p} x_{ij} = 1, \forall i = 1, \cdots, q$$

*For the bin-packing problem with $p = 3$ and $q = 2$, a corresponding bipartite is illustrated in figure 11, where the red node represents decision variables and the blue nodes represent constraints.*

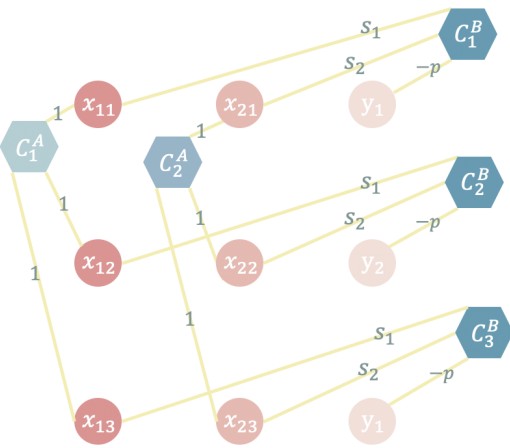

Figure 11: Bipartite for a bin-packing problem. Different colors indicate that the nodes are colored using the WL test. This figure illustrates the representation of a symmetric decomposable graph. There are four groups of nodes with the same colors in each group, and two nodes with distinct colors. In addition, a node in any group, for example, the lightest red group, only connects with one node in other groups.

*Such graphs are quite special since by excluding uniquely colored nodes and their connecting edges, the remaining symmetric nodes (nodes labeled in the same color via the WL test) can be combined to form several isomorphic, disconnected, and unfoldable graphs, as the dashed line highlights in Figure 12.*

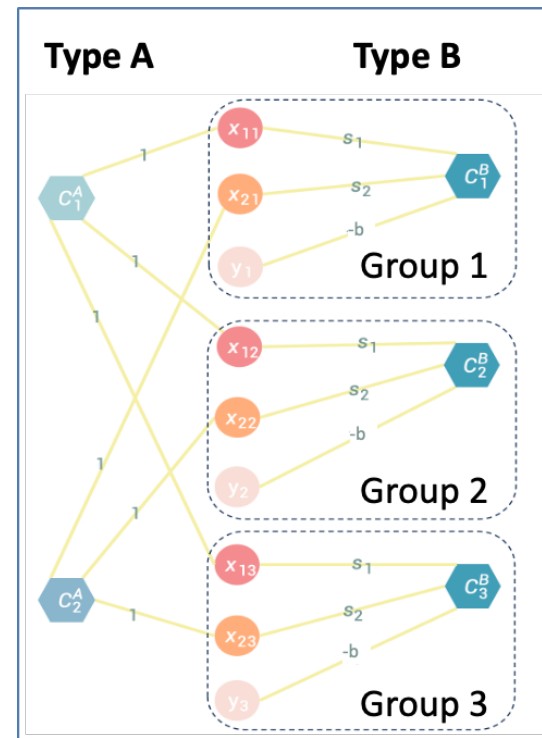

Figure 12: Decompose a symmetric decomposable graph for a bin-packing instance: After applying the WL test, the graph contains two types of nodes: **Type-A**: uniquely colored nodes $(C_1^A, C_2^A)$; **Type-B**: 3 groups of nodes that share the same color pattern. Each group contains nodes with distinct colors, the two groups have identical color distributions, and there are no edges between the groups.

**Example 8** (Symmetric Decomposable Problem: Set-Covering). *Consider a public transportation planning task in which a city aims to select a minimal set of bus stops such that all residential areas are served. Let $S$ denote the set of potential stop locations and $A$ the set of residential areas. The service relationship is encoded in a binary coverage matrix $C$, where $C_{ij} = 1$ if stop $j \in S$ can serve residential area $i \in A$, and $C_{ij} = 0$ otherwise. The objective is to choose the smallest subset of stops that collectively cover all residential areas.*

*To model this decision problem, we define a binary decision variable $y_j$ for each potential stop $j \in S$, where $y_j = 1$ indicates that stop $j$ is selected. The resulting optimization model is:*

$$\min_{y_j} \quad \sum_{j \in S} y_j$$

$$s.t. \quad \sum_{j \in S} C_{ij}\, y_j \geq 1, \qquad \forall i \in A,$$

$$y_j \in \{0, 1\}, \qquad \forall j \in S.$$

*Now, consider a small example with 4 potential stops and 4 residential areas. The set of potential stops is*

$$S = \{0, 1, 2, 3\},$$

*and the set of residential areas is*

$$A = \{0, 1, 2, 3\}.$$

*The coverage relationship is encoded in the binary matrix $C$, where $C_{ij} = 1$ indicates that stop $j$ covers area $i$. For this instance, the coverage matrix is*

$$C = \begin{pmatrix} 0 & 0 & 1 & 1 \\ 0 & 0 & 0 & 1 \\ 1 & 0 & 1 & 0 \\ 1 & 1 & 0 & 0 \end{pmatrix}.$$

*Each row corresponds to a residential area, and each column corresponds to a potential stop. For example, the first row indicates that area 0 is covered by stops 2 and 3, and the second row shows that area 1 is covered only by stop 3.*

*Substituting these parameters into the general set covering formulation yields the following optimization problem:*

$$\min_{y_j \in \{0,1\}} \quad y_0 + y_1 + y_2 + y_3$$

$$s.t. \quad y_2 + y_3 \geq 1,$$

$$y_3 \geq 1,$$

$$y_0 + y_2 \geq 1,$$

$$y_0 + y_1 \geq 1.$$

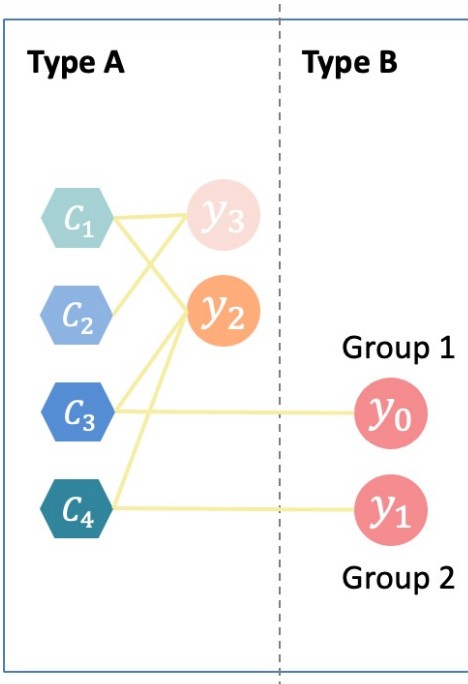

Figure 13: Decompose a symmetric graph for a set-covering instance: After applying the WL test, the graph contains two types of nodes: **Type-A**: uniquely colored nodes ($C_1, C_2, C_3, C_4, y_2, y_3$); **Type-B**: two groups of nodes that share the same color pattern. Each group contains nodes with distinct colors, the two groups have identical color distributions, and there are no edges between the groups.

*As illustrated by the corresponding bipartite graph in Figure 13, the set-covering instance is symmetrically decomposable.*

**Example 9** (Symmetric Decomposable Problem Instance: Capital Budgeting Problem). *Consider a capital budgeting problem in which a firm must decide which projects to invest in over multiple planning stages. Each project provides a Net Present Value (NPV) and requires a certain investment cost as well as multiple types of resources. Denote the set of candidate projects and $T$ the set of planning stages. The value of project $i \in I$ is given by $\mathrm{NPV}_i$, and its investment cost is denoted by $c_i$. Projects also consume resources: project $i$ uses $a_{ir}$ units of resource type $r$ out of a per-stage resource availability $R_{tr}$.*

*To model sequential investment decisions, we introduce binary decision variables $x_{it}$ indicating whether project $i$ is undertaken in stage $t$. The goal is to maximize the total NPV accumulated through investments across all stages. The optimization model is:*

$$\max_{x_{it}} \quad \sum_{t \in T} \sum_{i \in I} \text{NPV}_i \, x_{it}$$

$$s.t. \quad \sum_{i \in I} a_{ir} \, x_{it} \ \leq \ R_{tr}, \quad \forall t \in T, \ \forall r,$$

$$x_{it} \in \{0,1\}, \qquad \forall i \in I, \ \forall t \in T.$$

*Now, let*

- $I = \{1,2,3\}$ *be the index set of candidate projects.*

- $T = \{1,2\}$ *be the set of planning stages.*

- $\text{NPV}_i$ *denotes the net present value of project $i$. The objective coefficient for project $i$ in every stage is $\text{NPV}_i$.*

- $r_{ir}$ *denotes the consumption of resource type $r$ by project $i$. Here $r = 1,2$ indexes two distinct resource types.*

- $R_{tr}$ *denotes the availability (limit) of resource $r$ at stage $t$.*

- $x_{it} \in \{0,1\}$ *is a binary decision variable equal to one if project $i$ is undertaken in stage $t$, and zero otherwise.*

*Then we have an instance for capital budgeting problem:*

$$\max_{x_{it} \in \{0,1\}} \quad \sum_{t=1}^{2} \sum_{i=1}^{3} \text{NPV}_i \, x_{it}$$

$$s.t. \quad \sum_{i=1}^{3} r_{ir} \, x_{it} \leq R_{tr}, \qquad \forall t = 1,2, \forall r = 1,2,$$

$$x_{it} \in \{0,1\}, \qquad \forall i = 1,2,3, \forall t = 1,2.$$

*As illustrated by the corresponding bipartite graph in Figure 14, the capital budgeting instance is symmetrically decomposable.*

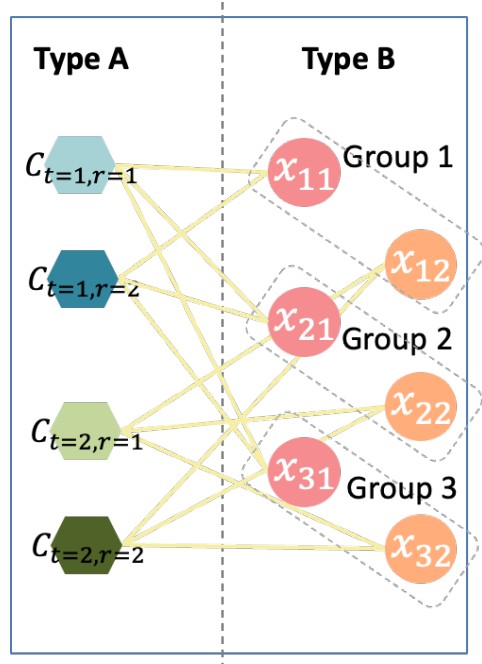

Figure 14: Decompose a symmetric graph for a capital budgeting instance: After applying the WL test, the graph contains two types of nodes: **Type-A**: uniquely colored nodes ($C_{t=1,r=1}, C_{t=1,r=2}, C_{t=2,r=1}, C_{t=2,r=2}$); **Type-B**: 3 groups of nodes that share the same color pattern. Each group contains nodes with distinct colors, the two groups have identical color distributions, and there are no edges between the groups.

**Example 10** (Solver can be Inconsistant).

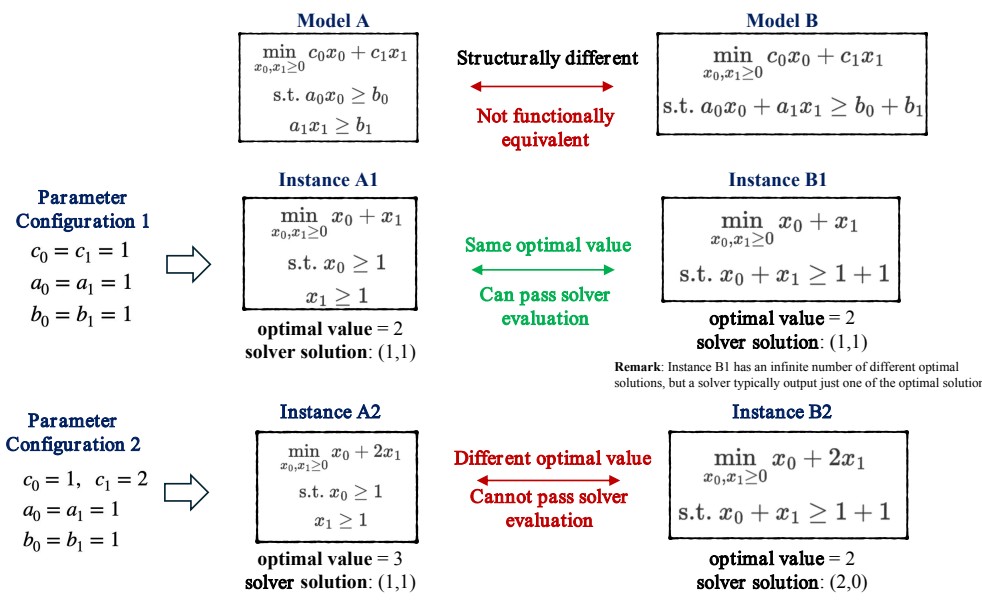

Figure 15: Solver's evaluation on modeling equivalence can be inconsistent across different parameter configurations

## F  ERROR ANALYSIS

Understanding how models fail is essential for interpreting benchmark performance and diagnosing model limitations. Therefore, we conducted an additional error analysis on three representative LLMs (GPT-4o, GPT-5, and O1) by annotating 1181 answers and categorizing them into two major groups: compile errors and modeling errors.

Compile Errors include issues visible directly from the execution logs, including

- Basic Python mistakes: indentation errors, syntax errors, etc.
- Gurobi-specific errors: incorrect function names, etc.
- Code errors stemming from a misunderstanding of the OR model: e.g., misunderstanding input formats or the meaning of variables.

Modeling Errors include mistakes in the optimization formulation:

- Incorrect objective functions
- Incorrect constraints
- Incorrect decision variables

Note that we report the full incidence of modeling error as they may co-occur. The distribution of error types (percentage of test errors attributable to each type) is listed in Table 6. As LLMs improve, compile errors related to basic coding abilities (Python/Gurobi) steadily decrease (O1 < GPT-5 < GPT-4o), while constraint-related mistakes dominate overall failures.

Table 6: Error analysis on Bench4Opt

| Error Type | GPT-4o | GPT-5 | O1 |
|---|---|---|---|
| Python understanding | 9.97 | 0.41 | 0.00 |
| Gurobi understanding | 5.30 | 0.41 | 0.40 |
| Modeling understanding | 4.36 | 2.48 | 1.61 |
| Wrong objective | 3.43 | 2.07 | 3.23 |
| Wrong constraints | 56.07 | 77.69 | 60.08 |
| Wrong variable | 27.41 | 32.23 | 39.92 |

