# OpenReview forum: "ORGEval: Graph-Theoretic Evaluation of LLMs in Optimization Modeling"
_ICLR.cc/2026/Conference — ICLR 2026 Conference Desk Rejected Submission_

### Official Review · Reviewer_NtBm · 2025-10-19

**Soundness:** 3
**Presentation:** 2
**Contribution:** 2
**Rating:** 4
**Confidence:** 3

**Summary:**

This paper focuses on evaluating whether LLMs can correctly formulate an optimization model. The authors propose a framework called ORGEval, which represents optimizations models as graphs and tests model equivalence via graph isomorphism rather than by solving instances. The authors prove a sufficient condition called symmetric decomposable (SD) under which a Weisfeiler–Lehman (WL) test is guaranteed to decide isomorphism. ORGEval first verifies SD and then applies a tailored WL test.

The authors also release a benchmark called Bench4Opt which has 394 problems where all reference instances satisfy the SD condition. Using ORGEval, evaluation is very efficient and 100% consistent across five random configurations, while solver-based comparisons fail frequently and can disagree.

**Strengths:**

The authors introduced a new evaluation paradigm that shifts from solver-based numerical checks to graph isomorphism structure evaluation. This shows better runtime efficiency and consistency compared to solver-based baselines on their benchmark. The Bench4Opt benchmark is the first model data separated benchmark for optimization modeling, and provides a valuable resource for evaluating LLM’s modeling capabilities.

**Weaknesses:**

1. The WL-based decision is guaranteed correct only when the symmetric decomposable (SD) condition holds, and all Bench4Opt problems happen to satisfy SD. This risks overestimating real-world coverage if SD is less common.
2. Equivalence over model–data separation is approximated by testing five random parameter draws, but this is under-justified.
3. The evaluation framework relies on strict graph isomorphism, and can unfairly penalize mathematically correct but structurally different formulations (it rewards structural replication but not genuine modeling understanding and alternative solutions).
4. The paper presents aggregate accuracy scores but lacks an analysis of the types of errors LLMs are making

**Questions:**

1. How prevalent do you think the SD property is in complex optimization problems
2. The paper tests model-level equivalence on only five random parameter configurations, but could two non-equivalent models coincidentally appear equivalent under such limited sampling?

---

> ### Author Response · Authors · 2025-11-28
> **In response to Weakness 1 and Question 1**
>
> > ... all Bench4Opt problems happen to satisfy SD. This risks overestimating real-world coverage if SD is less common.
>
> > How prevalent do you think the SD property is in complex optimization problems
>
> Thank you for this insightful question. From our empirical experiments, we found that the SD property is common in many real-world, complex optimization problems. Our conclusion is supported by the following evidence:
>
> **1. Validation on a Broad Set of LLM-Generated Classic Problems**
>
> We used GPT-4o to generate a diverse set of LP and MILP, including
>
> *Diet Problem, Transportation Problem, Blending Problem, Production Planning Problem, Network Flow Problem, Portfolio Optimization Problem, Cutting Stock Problem, Staff Scheduling Problem, Knapsack Problem, Traveling Salesman Problem, Vehicle Routing Problem, Bin Packing Problem, Set Covering Problem, Capacitated Facility Location Problem, Capital Budgeting Problem, and Assignment Problem*.
>
> We did not guide the LLM to generate instances with any specific structural property, nor did we selectively choose among its outputs. Nevertheless, all LLM-generated instances of these classic problems turned out to be symmetric-decomposable. This suggests that SD structure arises naturally in the formalization of many common, structured problems.
>
> **2. Empirical Analysis on the Symmetric TSPLib Dataset**
>
> To test the SD property on complex optimization problems, we systematically evaluated instances from the symmetric TSPLib (source: https://github.com/mastqe/tsplib). Due to the time limit, we verified the SD property on all instances with fewer than 500 cities (~ 500,000 graph nodes). All of these tested instances satisfied the SD property. This provides strong evidence of the property’s prevalence in this well-known and challenging problem class.
>
> **3. Additional Examples for SD in the Paper Appendix**
> To further aid the community's understanding, we have included several additional examples of SD problems and their corresponding colored bipartite graph in Appendix E.3,  Example 7.8.9
>
> In addition, we would like to offer some intuition besides the above empirical evidence. The key idea of SD is as following
>
> High symmetry in a graph generally makes graph isomorphism detection more difficult. However, assigning features or values to nodes and edges can break some of the symmetry, making equivalence checking easier. An SD graph can be decomposed into several subgraphs that are themselves non-symmetric. If a class of models allows random sampling of certain parameters, it becomes highly likely that an SD instance can be obtained. In practice, many model types naturally satisfy this assumption; therefore, we can easily construct SD instances via random parameter instantiation.
>
> In summary, we think the SD property is not a narrowly defined condition but is a common feature in many complex optimization problems.

---

> ### Author Response · Authors · 2025-11-28
> **In response to Weakness 2 and Question 2**
>
> > Equivalence over model–data separation is approximated by testing five random parameter draws, but this is under-justified.
>
> > The paper tests model-level equivalence on only five random parameter configurations, but could two non-equivalent models coincidentally appear equivalent under such limited sampling?
>
>
> Thank you for raising this valuable point. We agree that consistency on five random instances does not guarantee consistency over the entire parameter space. To strengthen this claim, we have added an additional experiment in the rebuttal.
>
> For each model in Bench4Opt, we randomly generated 50 instance pairs (a ground-truth instance and a tested instance, where the tested instance is produced from DeepSeek-V3’s model formulation). We then examined whether ORGEval and solver-based evaluation yield consistent equivalence judgments across these instances.
>
> - ORGEval achieves 100% consistency across all 50 instance pairs for every model.
> - Solver-based evaluation only successfully evaluates 30.23% of the models, i.e., those with both instances solved to optimal solutions. And the solver’s consistency across the 50 instances is only 87.18% amount these successfully evaluated models.
>
> Empirically, we do not observe any non-equivalent models being incorrectly judged as equivalent by ORGEval, whereas solver-based evaluation may produce such false positives for 12.83% of the models. We acknowledge that evaluating 50 instances per model does not fully characterize all possible data configurations. However, given the time constraints of the rebuttal period, this expanded experiment demonstrates that a single-instance ORGEval evaluation provides a much more reliable approximation of model-level equivalence than a single-instance solver-based evaluation.

---

> ### Author Response · Authors · 2025-11-28
> **In response to Weakness 3**
>
> > The evaluation framework relies on strict graph isomorphism, and can unfairly penalize mathematically correct but structurally different formulations.
>
> We appreciate the reviewer's attention to our framework. Our evaluation method is for modeling correctness detection in benchmarking. The goal for the OR benchmarking task is to verify that the LLM has translated the textual problem statement into a mathematical formulation. Under this setting, graph-structure equivalence is a reliable and appropriate test: structural isomorphism implies the formulations match at the variable/constraint level and therefore the translation is correct.
>
> There exist optimization problems with mathematically correct but structurally different formulations. For example, adding slack variables and rescaling do not change the solution for a model. However, such reformulations alter the structure and physical interpretation specified by the problem text. In our evaluation, a formulation that deviates from the textual description is considered incorrect, even if it preserves the optimal solution.
>
> We have clarified in our updated paper (after Definition 6) which types of equivalence ORGEval supports (renaming/permutation of variables and constraints) and which reformulations fall outside our evaluation scope.

---

> ### Author Response · Authors · 2025-11-28
> **In response to Weakness 4**
>
> > The paper presents aggregate accuracy scores but lacks an analysis of the types of errors LLMs are making.
>
> Thank you for the suggestion. We conducted an additional error analysis.  Due to the time limit, we categorized errors made by three representative LLMs (GPT-4o, GPT-5, and O1) into two major groups: **compile errors** and **modeling errors**.
>
> Compile Errors include issues visible directly from the execution logs, including
> - Basic Python mistakes: indentation errors, syntax errors, etc.
> - Gurobi-specific errors: incorrect function names, etc.
> - Code errors stemming from a misunderstanding of the OR model: e.g., misunderstanding input formats or the meaning of variables.
>
> Modeling Errors include mistakes in the optimization formulation:
> - Incorrect objective functions
> - Incorrect constraints
> - Incorrect decision variables
>
> Note that, modeling errors may co-occur, so we report their full incidence. The distribution of error types (percentage of test errors attributable to each type) is:
>
> | Error Type             | GPT-4o | GPT-5 | O1    |
> | ---------------------- | ------ | ----- | ----- |
> | Python understanding   | 9.97   | 0.41  | 0.00  |
> | Gurobi understanding   | 5.30   | 0.41  | 0.40  |
> | Modeling understanding | 4.36   | 2.48  | 1.61  |
> | Wrong objective        | 3.43   | 2.07  | 3.23  |
> | Wrong constraints      | 56.07  | 77.69 | 60.08 |
> | Wrong variable         | 27.41  | 32.23 | 39.92 |
>
>
>
> **Key Observations.**
> 1. As LLMs improve, compile errors related to basic coding abilities (Python/Gurobi) steadily decrease (O1 < GPT-5 < GPT-4o).
> 2. Modeling errors dominate overall failures, with constraint-related mistakes consistently representing the largest share.
>     This suggests that while modern LLMs can increasingly produce syntactically correct code, accurately capturing the structure of optimization models remains challenging.

---

### Official Review · Reviewer_WWVn · 2025-10-31

**Soundness:** 3
**Presentation:** 2
**Contribution:** 3
**Rating:** 4
**Confidence:** 4

**Summary:**

This paper proposes the ORGEval, which represents optimization models as graphs, reducing equivalence detection to graph isomorphism testing.

**Strengths:**

This paper innovatively uses graph to determine whether the optimization problem modeling is correct.

**Weaknesses:**

- Is it reliable to judge the correctness of a model by examining its graph structure? Are there complex optimization problems where different modeling methods exist, but the final result is always correct? For example, some problems in combinatorial optimization, or problems with duality.
- For complex problems, correct modeling and correct solution are not equivalent. For example, some complex problems may have time complexity issues when solving them, and simple modeling methods may prevent the problem from being solved within the limited time and space resources. Optimization problems that use complex modeling techniques are not necessarily isomorphic to the graph given by the ground truth.
- What are the problem sizes covered by ORGEval? For example, as shown in Appendix B.1, what is the approximate distribution of the number of variables and constraints for each type of problem? For instance, how many nodes are there in the Traveling Salesman Problem?
- In modeling certain types of problems, the constraints don't seem to increase with the complexity of the data. For example, in the Traveling Salesman Problem, regardless of the number of nodes, the constraints don't appear to become more complex, so the complexity of modeling this problem doesn't increase. In this case, the types of problems covered by the benchmark are more important than the number of problems. How does this article view and consider this issue?
- How do the chat model (e.g. deepseek-v3) and the thinging model (e.g. deepseek-r1) perform differently on this type of task? Could you provide a detailed analysis?

**Questions:**

As described in weaknesses.

---

> ### Author Response · Authors · 2025-11-28
> **In response to Weakness 1 and 2**
>
> We appreciate the reviewer’s attention to the scope of our evaluation setting.
>
> > Is graph-structure checking reliable?
>
> In our setting, graph-structure checking is reliable for modeling correctness detection in benchmarking. The goal for the OR benchmarking task is to verify that the LLM has translated the textual problem statement into a mathematical formulation. Under this setting, graph-structure equivalence is a reliable and appropriate test: structural isomorphism implies the formulations match at the variable/constraint level and therefore the translation is correct.
>
> > Different modeling methods exist, but the final result is always correct
>
> There exist optimization problems where different modeling methods exist, while the final result is always correct. For example, adding slack variables and rescaling do not change the solution for a model. However, such reformulations alter the structure and physical interpretation specified by the problem text. In our evaluation, a formulation that deviates from the textual description is considered incorrect, even if it preserves the optimal solution.
>
> > For complex problems, correct modeling and correct solution are not equivalent. ... Optimization problems that use complex modeling techniques are not necessarily isomorphic to the graph given by the ground truth."
>
> We agree that, for complex problems, correct modeling and correct solution are not equivalent, and certain solution-preserving reformulations may not be isomorphic to the ground-truth graph. Our work does not aim to detect all such forms of solution-preserving equivalence. Again, our goal is to verify that the LLM has translated the textual problem statement into a mathematical formulation. Under this setting, structural isomorphism implies whether the translation is correct.
>
> We have clearly clarified in our paper  (see our clarification of evaluation principles after Definition 6) which types of equivalence ORGEval supports (renaming/permutation of variables and constraints) and which reformulations fall outside our evaluation scope.

---

> ### Author Response · Authors · 2025-11-28
> **In response to Weakness 3**
>
> > What are the problem sizes covered by ORGEval? For example, as shown in Appendix B.1, what is the approximate distribution of the number of variables and constraints for each type of problem? For instance, how many nodes are there in the Traveling Salesman Problem?
>
> Thank you for the question. As there is no universally accepted or rigorous taxonomy for mapping each optimization instance to a single problem class, it is difficult to assign every individual problem in our benchmark to a unique class without introducing arbitrariness. For this reason, instead of providing a per-instance categorization, we report the overall distributions of variables and constraints across the entire benchmark.
>
> As shown in Appendix B.3 Figure 4, although the majority of instances contain fewer than 50 variables and constraints, the benchmark also includes a nontrivial number of instances with more than 200 variables or constraints, which is substantially larger and more complex than those in existing benchmarks.

---

> ### Author Response · Authors · 2025-11-28
> **In response to Weakness 4**
>
> > In modeling certain types of problems, the constraints don't seem to increase with the complexity of the data. For example, in the Traveling Salesman Problem, regardless of the number of nodes, the constraints don't appear to become more complex, so the complexity of modeling this problem doesn't increase. In this case, the types of problems covered by the benchmark are more important than the number of problems. How does this article view and consider this issue?
>
> Thank you for raising this important point. We fully agree that for certain optimization tasks, such as the Traveling Salesman Problem, the underlying model structure does not inherently grow in complexity as the data scale increases. Consequently, when constructing Bench4Opt, our primary focus was **not** on increasing the number of problem instances but rather on **broadening the diversity of problem types/domains and capturing structural variability across models**.
>
> To address this, we carefully designed a data-generation pipeline (detailed in Appendix B). Here we summarize several key design choices:
>
> 1.  **Model–data separation.** We adopt a problem format that explicitly decouples model parameters from the problem description. This allows us to control the complexity and diversity of the dataset independent of instance size.
>
> 2.  **Seed-to-variant augmentation.** For each problem class, we begin with a classical “seed” model and generate structurally meaningful variants. For example, in bin packing, we prompt GPT-4o to propose variants on constraints, objectives, or variables. To reduce redundancy, the prompt includes a concise summary of all historical variant augmentation.
>
> 3.  **Broad coverage across classic and practical domains.** As a result of this process, Bench4Opt covers over 15 classical seed optimization problems and more than 80 practical domains derived from them. Detailed statistics and coverage are provided in Appendix B.3.
>
> We hope this clarifies our perspective, in settings where structural complexity does not scale with data size, diversity of problem types, and model structures is indeed the key consideration, and Bench4Opt is designed explicitly with this principle in mind.

---

> ### Author Response · Authors · 2025-11-28
> **In response to Weakness 5**
>
> > How do the chat model (e.g. deepseek-v3) and the thinging model (e.g. deepseek-r1) perform differently on this type of task? Could you provide a detailed analysis?
>
> Thank you for the suggestion. We conducted an additional error analysis. Due to the time limit, we categorized errors made by three representative LLMs (GPT-4o, and O1) into two major groups: **compile errors** and **modeling errors**.
>
> Compile Errors  include issues visible directly from the execution logs, including
>
> - Basic Python mistakes: indentation errors, syntax errors, etc.
>
> - Gurobi-specific errors: incorrect function names, etc.
>
> - Code errors stemming from a misunderstanding of the OR model: e.g., misunderstanding input formats or the meaning of variables.
>
> Modeling Errors include mistakes in the optimization formulation:
>
> - Incorrect objective functions
>
> - Incorrect constraints
>
> - Incorrect decision variables
>
> Importantly, modeling errors may co-occur, so we report their full incidence. The distribution of error types (percentage of test errors attributable to each type) is:
>
> | Error Type | GPT-4o | O1 |
> | ---------------------- | ------ | ----- |
> | Python understanding | 9.97 | 0.00 |
> | Gurobi understanding | 5.30 | 0.40 |
> | Modeling understanding | 4.36 | 1.61 |
> | Wrong objective | 3.43 | 3.23 |
> | Wrong constraints | 56.07 | 60.08 |
> | Wrong variable | 27.41 | 39.92 |
>
> **Key Observations**
>
> 1.  Code-generation ability improves in the reasoning model. O1 essentially eliminates all compile errors related to basic Python coding abilities and nearly all errors due to Gurobi coding errors, performing better than GPT-4o. This suggests that O1’s structured reasoning process leads to more robust execution-ready code.
>
> 2.  Modeling-understanding ability remains. While their distributions differ slightly, all three models show comparable difficulty in producing correct optimization formulations, and Constraint-related errors dominate for every model.

---

### Official Review · Reviewer_kxC7 · 2025-10-31

**Soundness:** 1
**Presentation:** 2
**Contribution:** 1
**Rating:** 2
**Confidence:** 4

**Summary:**

The paper studies the translating problem descriptions into optimization models using LLMs from the angle of verification of results.

The main claim is that, instead of relying on ground truth optimal values, the proposed ORGEval represents MIPs as a standard weighted bipartite graphs and reduces equivalence detection to a graph isomorphism problem between the LLM generated model vs. ground truth model. It leverages the Weisfeiler–Lehman (WL) test under a newly proposed sufficient condition called symmetric decomposable (SD), ensuring correctness when this condition holds.

The paper also presents Bench4Opt, a dataset of optimization problem descriptions, models and instances, and LLM model generation accuracy on this dataset using a suite of different recent LLMs.

**Strengths:**

Interesting and relevant topic for using LLM in translation task for optimization problems.
Introduction of a new dataset is always a great contribution to the community
There is rigor to the new SD condition

**Weaknesses:**

I liked reading this paper but fail to see its significance.

At a high-level, the paper moves the assumption around from knowing the optimality to knowing the model. Still, this requires known a ground truth model --which is arguably is the most daunting part of verification.

The difficult part in model translation is having the ground truth; either in the form of a model OR the optimal value that it produces. Only after, either of them of is known, one can check the correctness. IF the ground truth optimal value is known; equivalence check comes down to running the solver and a simple value comparison. Alternatively, IF the ground truth model is known; equivalence check comes down to running a form of graph isomorphism (this paper). In that lens; the paper does not change the difficulty of verifying LLM generated models, just changes what we verify. That's the major weakness of the paper.

Experiments are far from ideal:

The paper claims that running the solver might take time. Alternatively, one might claim that running several smaller instances already serve as a good proxy. One does not need to solve the hardest instance of a problem to verify a model (as done in the MIPLIB hard experiments which I don't see what Table 1 conveys).

On the contrary, running the solver might be an easier approach (since these solvers are quite established) as opposed to running an ad-hoc mechanism, like the one presented here. Btw, I did not see code/data links in the submission. Given the centrality of SD detection and WL implementation choices open‑sourcing is critical for this part. Same comment for the benchmark dataset as well.

Table 1 compares MIP solver runtime to prove optimality compared to an implementation of a checking algorithm. I don't see what this comparison achieves. This can be made arbitrarily positive in favor of the deterministic/poly algorithm here as to infinity for sizes above solver limits. Again, one does not need to solve the largest instances to check models. Aren't there other graph isomorphism based strong baselines to compare against where you can show the added value of your new SD proposal?

A better comparison to consider; set the time-limit to the runtime of your algorithms and then check if there NO instances that a solver can find the ground truth optimal value. IF that's the case, now your method is providing a verification that was not possible before. Still, the comparison is not apples-to-apples: solving MILPs is fundamentally harder than computing a WL coloring; evaluation cost differs in kind, not degree.

Does your method require the same number of variables and/or constraints? I don't think so, but it would be good to clarify in the paper. But I believe it requires that same parameter names are used, which is a strong assumption, no? Please discuss in the paper. Also, what happens when there are slack variables or constraint aggregations (common in OR models) that still preserves model correctness? Would your equivalence check work with that?

There is an assumption in the overall algorithm that "all ground truth instances are selected to be symmetric". That greatly simplifies the equivalence checking, no? How can all the ground truth guaranteed to be symmetric? Please explain this further

I am not following what's Table 2 is trying to provide. Why would anyone run models with random parameters? IF all instances in this dataset are somehow SD, isn't it a tautology that your method would return %100.

What would have happened if your dataset had non-SD instances?

Table 3, the performance of current LLMs modeling these problems is a nice addition (it's good to know) but not related to the paper's main claims. Btw, in this table, what is structured vs. unstructured optimization tasks. This was never defined, so I am not even sure what this table shows. Also, are this running LLMs one shot against the dataset. There are advanced modeling co-pilots so it would be more interesting to see their results on this dataset rather than vanilla prompting (but again, this is not the main part of the paper so it's ok)

Regarding existing work, a lot of improvements are needed.
- The similarity/difference between this and the existing work such as EquivaMap is never discussed in the main paper. This is a major problem and must be fixed.

- The claim that Bench4Opt is "the first model-data separated dataset for optimization modeling" is NOT correct. Please see Text2Zinc: A Cross-Domain Dataset for Modeling Optimization and Satisfaction Problems in MiniZinchttps://arxiv.org/abs/2503.10642. Text2Zinc must be cited and this claim must be revised. In fact, it is important to point out that this new dataset still hardcodes the parameter names to be found in the data as part of the textual description as well as the variable names and their types. I appreciate the effort but that's still not "true" natural textual representation of optimization problems. That does not take away from the hard work and the effort of the team in curating and verifying --which is crucial for the community. Thank you!

- Regarding LLM modeling co-pilots, consider also covering Ner4Opt (Constraint'24, https://link.springer.com/article/10.1007/s10601-024-09376-5), ChatOpt(CP'24 https://drops.dagstuhl.de/entities/document/10.4230/LIPIcs.CP.2024.20) and OptiChat (https://arxiv.org/abs/2501.08406). These would help noting that "asserting optimal values against the ground truth" is NOT the only verification that exists in the prior work, especially for Feasibility/Satisfaction/Logic problems "asserting feasibility" is commonly done which must be mentioned.

- The section on Related Work in the Appendix would benefit from including earlier works, such as for algorithm configuration part: ISAC-Instance specific algorithm configuration (ECAI'10), and for algorithm selection part: Algorithm selection and scheduling (CP'12).

- Typos in references; like "consingh2012overviewuration" plz fix. Other typos: “isomporphism,” “sturctured,” “desctiption,” “euqivalent,” “choosen,” etc; plz spell check

- The formal definition of DS requires groups to be disconnected. But isn't the examples in Figure 10-11 has cross-group connections?

- Why would random parameter sampling yield SD instances with probability 1? that sounds too strong, no? Is it because we start with a dataset that's SD already?

- Complexity is claimed O(k(m+n)2) in main text and O(kmn) later. Why?

**Questions:**

See above

---

> ### Author Response · Authors · 2025-11-28
> **Clarify the Scope: ORGEval is a benchmarking tool, not an online verifier**
>
> > At a high-level, the paper moves the assumption around from knowing the optimality to knowing the model. Still, this requires known a ground truth model --which is arguably is the most daunting part of verification.
>
> > The difficult part in model translation is having the ground truth; either in the form of a model OR the optimal value that it produces. Only after, either of them of is known, one can check the correctness. IF the ground truth optimal value is known; equivalence check comes down to running the solver and a simple value comparison. Alternatively, IF the ground truth model is known; equivalence check comes down to running a form of graph isomorphism (this paper). In that lens; the paper does not change the difficulty of verifying LLM generated models, just changes what we verify. That's the major weakness of the paper.
>
> Thanks for the concern. We agree that our evaluation is based on a reference ground-truth. But we would like to clarify that **our goal is not to build an online verifier for real-world deployment, nor to eliminate the need for a ground-truth model**.
>
> ## We focus on benchmarking LLMs
>
> ORGEval is designed as a benchmarking framework for model-translation tasks in optimization. While some open-ended benchmarks (e.g., creative generation or free-form reasoning) can rely on human judgment or heuristic scoring, optimization modeling is a highly structured task, and evaluating whether a generated model faithfully captures the intended mathematical formulation benefits greatly from having a reference model.
>
> Existing work on evaluating LLM-generated optimization models typically relies on some form of reference signal—most commonly the optimal objective value obtained by solving the ground-truth instance.**ORGEval follows this general evaluation paradigm, but extends it to a stricter and more informative setting** by using the full reference formulation to assess structural correctness, rather than relying solely on optimal value comparison. Our contribution is to provide a more structured, parameter-stable, and scalable way to evaluate LLM-generated optimization models offline, rather than a tool for verifying models in real-time settings where the ground truth is unknown.
>
> ## Why benchmarking is important
>
> Beyond this clarification, we would like to emphasize **why benchmarking itself is a critical research direction:**
> 1. Reliable LLM benchmarks are indispensable for progress. Without stable and well-defined benchmarks, it becomes impossible to measure learning progress, compare methods fairly, or understand failure modes—especially as models become larger and more capable. In fields such as vision and NLP, breakthroughs were closely tied to the availability of robust benchmarks. OR modeling currently lacks such infrastructure, which constrains scientific progress.
>
> 2. Optimization modeling is a domain where benchmarking is particularly challenging and highly valuable, which has the following features: **difficult** (high combinatorial and structural complexity), **open-ended** (many ways to model the same problem), **underspecified** (textual descriptions often omit modeling conventions), and **parameter-sensitive** (model correctness may not be inferred from a single instance).
> This makes evaluation fundamentally harder than supervised tasks, symbolic tasks, or even code generation. As a result, a dedicated benchmark—together with a principled evaluation methodology—is vital for enabling systematic study.
>
> ## Additional benefit of our method
> Our method additionally supports outcome-based training. Because ORGEval provides a deterministic and structure-level correctness signal, it can be used not only for offline evaluation but also as a reward function in RL-based fine-tuning or other outcome-based training pipelines. In contrast, optimal-value comparison requires solving each instance, which may fail or require careful parameter tuning to ensure feasibility and reasonable runtime. ORGEval’s reward signal is solver-free and stable across instances, making it suitable for training LLMs to improve their capability in OR modeling.

---

> ### Author Response · Authors · 2025-11-28
> **Suggestion to use small instances to make solver evaluation faster**
>
> > The paper claims that running the solver might take time. Alternatively, one might claim that running several smaller instances already serve as a good proxy. One does not need to solve the hardest instance of a problem to verify a model (as done in the MIPLIB hard experiments which I don't see what Table 1 conveys).
>
> We appreciated the reviewer's suggestion to use multiple small solver runs instead of our method. This strategy, however, is not a reliable general-purpose verification approach for the following reasons:
>
> -  Some optimization problems remain slow to solve even at small sizes (For example,markshare2 is a hard problem in MIPLIB with only 74 variables and 7 constraints; See example in [https://anonymous.4open.science/r/ORGEval-Rebuttal-ICLR2026-009D/example_solver_slow/markshare2.lp](https://anonymous.4open.science/r/ORGEval-Rebuttal-ICLR2026-009D/example_solver_slow/markshare2.lp)).
> - Some problem classes cannot be validated by solver outcomes at all: For feasibility problems, the objective value is always a constant, comparison between objective values is non-informative (see example 3 in Appendix E). Small instances are not informative either.
> - Solver outcomes across small instances can be inconsistent: A model may appear correct under some parameter choices but incorrect under others; when these outcomes disagree, there is no principled way to decide which result to trust. (See example 7 in Appendix E and Table 2)
>
> > running the solver might be an easier approach (since these solvers are quite established) as opposed to running an ad-hoc mechanism, like the one presented here. Btw, I did not see code/data links in the submission.
>
> Our code and data are included in the “Supplementary Material” ZIP file attached to the submission. Specifically, the package contains:
> -  Implementation of ORGEval, including our SD-detection routine and ORGEval pipeline.
> - The full Bench4Opt dataset, including word problems, ground truth model in gurobi code format, instance data files, and model instances.
>
> We will also publicly release the repository upon acceptance to ensure full reproducibility and transparency.

---

> ### Author Response · Authors · 2025-11-28
> **Experiment in table 1 and table 2**
>
> > Table 1 compares MIP solver runtime to prove optimality compared to an implementation of a checking algorithm. I don't see what this comparison achieves. This can be made arbitrarily positive in favor of the deterministic/poly algorithm here as to infinity for sizes above solver limits.
>
> We appreciate the reviewer's comment, but believe there is a fundamental misunderstanding about the purpose of this comparison. The table is **not** intended to show that our algorithm is "faster" than MIP solvers on easily solvable problems—that would indeed be a meaningless comparison.
>
> Instead, the key point is this: **SOTA solver cannot find a solution does not imply that the problem is unrealistic.** Our method and experiment in Table 1 addressed exactly this gap and provide rigorous verification for problems that are intractable for SOTA solvers.
>
> The MIPLIB hard instances we selected represent genuine modeling challenges that occur in practical applications. The fact that commercial solvers cannot solve them within a reasonable time doesn't make these problems "unrealistic"—it makes them representative of the complex optimization challenges faced in industry today.
>
> We provide a polynomial-time alternative when the solver fails. This isn't an "arbitrary" advantage—it's a practical solution for real-world problem classes where traditional approaches hit fundamental computational barriers.
>
> > I am not following what's Table 2 is trying to provide. Why would anyone run models with random parameters? IF all instances in this dataset are somehow SD, isn't it a tautology that your method would return %100.
>
> Thanks for the reviewer's question. We have added more descriptions for Table 2 in the caption. The experiment in Table 2 aims to illustrate the following question:
>
> If we use the evaluation result in one model instance, to what extent can we make the same conclusion on another model instance under the same model?
>
> For each Bench4OPT problem, we evaluate LLM-generated formulations using both the Solver-based method and ORGEval over 5 randomly instantiated data instances. The table reports the proportion of instances for which the two evaluation schemes yield consistent results. We represent the following 3 metrics:
>
> - **ORGEval consistency** denotes the percentage of problems whose five instances have a consistent evaluation result under ORGEval.
> - **Solver consistency** denotes the percentage of problems whose five instances have consistent evaluation results under solver-based optimal value matching.
> - **Feasibility consistency** denotes the percentage of problems whose five instances can all be solved by the solver.

---

> ### Author Response · Authors · 2025-11-28
> **On Graph Isomorphism Testing and SD condition**
>
> > "Aren't there other graph isomorphism based strong baselines to compare against where you can show the added value of your new SD proposal?"
>
> The reviewer asks whether there are stronger graph-isomorphism–based methods that could serve as baselines for model-equivalence checking. To the best of our knowledge, no such stronger GI-based baselines are available for this specific task. Our conclusion is based on the following points:
>
> **No polynomial-time baseline for general graph**
> - Graph-isomorphism–based model equivalence checking requires identifying a permutation of variables and constraints that transforms Model A into Model B. In the worst case, a naive algorithm would need to consider all possible permutations of the node set, which is factorial in size. Although existing (graph isomorphism)GI algorithms typically use sophisticated pruning to avoid enumerating all permutations, the graph isomorphism problem still has no known polynomial-time algorithm in general (We also discussed this in section 3.2 - isomorphism testing paragraph).
>
> **Existing GI tools have limitations for our setting**
> - **NetworkX GI.**
> NetworkX provides a widely used implementation of VF2-style backtracking GI with support for node/edge attribute matching. We used it as an applicable baseline for featured graphs.
> We randomly selected a highly symmetric optimization problem, generated 100 random instances, and tested each instance against itself using nx.is_isomorphic. Even in this easiest setting, NetworkX was ≈15× slower than ORGEval due to the large permutation search space.
> - **Nauty/Traces and Bliss (not applicable).**
> These are state-of-the-art tools for unlabeled or color-labeled graph isomorphism, offering excellent practical performance. However, they do not natively support the rich node/edge feature structures required by optimization-model graphs (e.g., variable types, constraint senses, coefficients).
> While they allow colors, this is insufficient for encoding the full attribute matrix. Moreover, any limitation that prevents WL from resolving isomorphism on a featured graph also prevents these color-only GI tools from resolving it, because they rely on similar coloring refinement mechanisms.
>
> While tools like NetworkX, Nauty/Traces, and Bliss exist, none provides (i) polynomial-time guarantees, (ii) support for rich optimization-model features, or (iii) correctness guarantees for this domain. ORGEval fills this gap by combining WL testing with our novel SD condition and detection algorithm.

---

> ### Author Response · Authors · 2025-11-28
> **On Graph Isomorphism Testing and SD condition - continue**
>
> > Does your method require the same number of variables and/or constraints?
>
> ORGEval requires the **same number** of variables and constraints, but **does not require matching parameter, variable, or constraint names**. Slack variables or other structural reformulations are not supported, as our evaluation strictly follows the textual problem description; model equivalence only allows renaming and permutation of variables/constraints, which is stricter than conventional OR equivalence (see our clarification of evaluation principles after Definition 6).
>
> > There is an assumption in the overall algorithm that "all ground truth instances are selected to be symmetric decomposable". That greatly simplifies the equivalence checking, no? How can all the ground truth be guaranteed to be symmetric?
>
> We clarify the following points regarding the symmetric decomposable (SD) condition in our benchmark:
>
> 1.  **Detection algorithm**: We have a formal SD detection algorithm that identifies instances satisfying the sufficient condition, which we use to select models containing SD instances for benchmarking.
>
> 2.  **Purpose of SD filtering**: We select models that can generate SD instances via parameter instantiation, ensuring that equivalence checking can be correctly evaluated (as guaranteed by Theorem 3.1).
>
> 3.  **No over-simplification**: Requiring SD instances does not simplify equivalence checking. We first detect whether an instance is SD before performing equivalence checking. In our experiments, SD is easily satisfied: we found that all seed models proposed by LLMs naturally contain SD instances, allowing meaningful evaluation of structure-level equivalence.
>
> 4. **Requiring SD instances is not a strong assumption**. We have a formal detection algorithm, and we have theoretical guarantees that for many models, we can sample SD model instances with high probability. In practice, we can sample and filter enough instances to ensure the benchmark contains sufficient SD cases for meaningful evaluation.
>
> > Why would random parameter sampling yield SD instances with probability 1? That sounds too strong, no? Is it because we start with a dataset that's SD already
>
> The statement “with probability 1” holds only under certain mild assumptions (Theorems D.5 and D.6), which are easily satisfied in practice. It is not because the dataset is SD by construction.
>
> **Intuition**: High symmetry in a graph generally makes graph isomorphism detection more difficult. However, assigning features or values to nodes and edges can break some of the symmetry, making equivalence checking easier. An SD graph can be decomposed into several subgraphs that are themselves non-symmetric. If a class of models allows random sampling of certain parameters, it becomes highly likely that an SD instance can be obtained. In practice, many model types naturally satisfy this assumption; therefore, we can easily construct SD instances via random parameter instantiation.
>
> > The formal definition of DS requires groups to be disconnected. But aren't the examples in Figure 10-11 have cross-group connections?
>
> The definition of SD requires that after removing uniquely colored nodes, the remaining nodes can be partitioned into disconnected groups. In our example, once the uniquely colored nodes are excluded, the remaining subgraphs indeed become disconnected. Therefore, these examples fully satisfy the SD definition.
>
> > What would have happened if your dataset had non-SD instances?
>
> Our method is designed for LLM benchmarking and training evaluation, where we ensure that all ground-truth instances satisfy the SD sufficient condition using our detection algorithm. In this setting, if an LLM generates a model that is not SD, ORGEval correctly flags it as an incorrect formulation.

---

> ### Author Response · Authors · 2025-11-28
> **Benchmark dataset and benchmark experiment**
>
> > what is structured vs. unstructured optimization tasks... are this running LLMs one shot against the dataset. There are advanced modeling co-pilots...
>
> Thank you for the thoughtful comments. We agree that the main contribution of our paper is the proposed evaluation methodology, and the LLM benchmarking results are intended primarily as an application of this framework rather than a core claim. We would like to make a few clarifications regarding reviewers' comments.
>
> **Definition of Structured vs. Unstructured Optimization Tasks.**
>
> In Bench4OPT, the natural-language description of each problem is provided at two different levels of abstraction:
> - **Structured**:Following the conventions of INFORMS optimization modeling competitions, we design a problem template in which each instance contains explicit and well-organized sections, including background, problem description, parameters, objective, constraints/decision variables, implementation notes, and expected output format.
> - **Unstructured**: For this version, we merge and rewrite all descriptive sections (except the explicit parameter descriptions) into one paragraph. Compared with the structured setting, this version removes the information hierarchy and formatting cues, making the modeling task more complex.
>
> Figures 4 and 5 in the paper present examples of a structured and unstructured problem description.
>
> **Evaluation Protocol**
>
> For a fair comparison across models, all LLMs are evaluated in a pass@1 setting. The goal of this experiment is to measure the baseline OR-modeling capability across different LLMs, rather than to compare prompt-engineering strategies or agent-based frameworks.
>
> We agree that examining advanced modeling copilots or agentic approaches would be interesting. Since this lies slightly outside the central scope of the paper, we kept our main experiments focused on the core evaluation method. However, we plan to include additional results with agent-based methods (e.g., chain-of-experts pipelines) in the camera-ready version to enrich the analysis. We appreciate the reviewer’s constructive feedback and believe these clarifications will help strengthen the presentation.

---

> ### Author Response · Authors · 2025-11-28
> **On complexity of ORGEval**
>
> > Complexity is claimed $O(k(m+n)^2)$ in main text and $O(kmn)$ later. Why?
>
> We thank the reviewer for catching this inconsistency. The earlier $O(k(m+n)^2)$ bound was a naive estimate written before the algorithm was finalized. After completing Algorithm 3 and analyzing its operations, the actual complexity is $O(kmn)$. We overlooked updating the earlier statement, and we appreciate the reviewer for pointing this out.

---

> ### Author Response · Authors · 2025-11-28
> **In response to recommended related works and minor suggestions**
>
> Thanks the reviewer for helpful suggestions. We have corrected the typos and added the papers you recommended to the related work section. For EquivaMap, we have added a comparison between ORGEval and EquivaMap in the introduction section of the revised manuscript. Following the related works recommended by the reviewer, we found 2 additional relevant works and have included them as well. The added list of citations is listed below. We appreciate your feedback for helping us improve the completeness.
> - Singirikonda, Akash, Serdar Kadioglu, and Karthik Uppuluri. "Text2Zinc: A Cross-Domain Dataset for Modeling Optimization and Satisfaction Problems in MiniZinc." _arXiv preprint arXiv:2503.10642_ (2025)
> - Dakle, Parag Pravin, et al. "Ner4opt: Named entity recognition for optimization modelling from natural language." _International Conference on Integration of Constraint Programming, Artificial Intelligence, and Operations Research_. Cham: Springer Nature Switzerland, 2023.
> - Michailidis, Kostis, Dimos Tsouros, and Tias Guns. "Constraint modelling with LLMs using in-context learning." _30th International conference on principles and practice of constraint programming_. （2024.)
> - Chen, Hao, et al. "OptiChat: Bridging Optimization Models and Practitioners with Large Language Models." _INFORMS Journal on Data Science_ (2025).
> - Kadioglu, Serdar, et al. "ISAC–instance-specific algorithm configuration." _ECAI 2010_. Ios Press, 2010. 751-756.
> - Kadioglu, Serdar, et al. "Algorithm selection and scheduling." _International conference on principles and practice of constraint programming_. Berlin, Heidelberg: Springer Berlin Heidelberg, 2011.
> - Cai, Junyang, Serdar Kadioglu, and Bistra Dilkina. "Gala: Global LLM Agents for Text-to-Model Translation." _arXiv preprint arXiv:2509.08970_ (2025).
> -Tsouros, Dimos, et al. "Holy grail 2.0: From natural language to constraint models." _arXiv preprint arXiv:2308.01589_ (2023).

---

### Official Review · Reviewer_NuHH · 2025-11-01

**Soundness:** 2
**Presentation:** 2
**Contribution:** 1
**Rating:** 4
**Confidence:** 3

**Summary:**

The author proposes a framework to assess whether LLM-generated optimization models were equivalent (to reference model or another generated model) by representing model instances as bipartite graphs and reducing the problem as a graph isomorphism problem. The core contribution is identifying symmetric decomposable (SD) graphs as a sufficient condition (instead of unfoldable) under which the Weisfeiler–Lehman (WL) test is guaranteed to correctly detect isomorphism. The authors also introduce Bench4Opt, a dataset of 394 optimization problems with model-data separation, and use it to benchmark several state-of-the-art LLMs.

**Strengths:**

- Authors made a fair case that current solvers are not efficient or consistent in identifying equivalence in optimization problems, due to difficulty in handling different parameter configurations, computational costs, and inability to handle infeasible instances
- Experiments demonstrated that their methodology works for this subclass of problems, with 100% consistency across random parameter configurations
- The theoretical characterization of SD graphs and proof that WL-test correctly determines isomorphism under this condition appears sound

**Weaknesses:**

- Limited contribution. Essentially, the authors added a check for symmetric decomposable on top of a Weisfeiler–Lehman (WL) test to check whether two optimization models are equivalent. The core pipeline uses the graph representation and WL-test from previous work, and the addition of SD detection being the only novel algorithmic component.
- Limited testing on other datasets, only worked on its own benchmark.
- The benchmark dataset is also relatvely small (394), and selection / dataset construction process is not clearly described beyond a list of sub-classes of problems.s
- Limited explanation on testing performacne on different LLMs on the benchmark. e.g.
- Accuracy is binary (equivalent v.s. not-equivalent) - Would be interesting to see the degree to which a model is inaccurate and also how (e.g. is it a matter of unable to translate the requirements?)

**Questions:**

- Is it possible to have non-SD instances that are equivalent? Does the framework just reject them altogehter?
- How are the "hand-crafted" problems derived? There are >100,000 entreis in MIPLIB, how is the Bench4Opt subset selected?
- Is there a dataset breakdown of the benchmark's composition?
- For the dataset, the paper mentions "two level of abstract, structured and unstructured description" -  How do they differ? and which one is fed into the model?

---

> ### Author Response · Authors · 2025-11-28
> **In response to Weakness 1**
>
> > Limited contribution. Essentially, the authors added a check for symmetric decomposable on top of a Weisfeiler–Lehman (WL) test to check whether two optimization models are equivalent. The core pipeline uses the graph representation and WL-test from previous work, and the addition of SD detection is the only novel algorithmic component.
>
> We thank the reviewer for acknowledging the contribution of SD detection. However, the contribution of this work goes substantially beyond “addition of SD detection” on top of an existing pipeline. Our novelty lies in three complementary dimensions:
>
> **1. Methodological novelty: a new formalization of model equivalence under model–data separation.**
>
> Existing solver-based equivalence tests are unreliable and fundamentally incompatible with the model–data separated setting that is common in real applications. We are the first to formalize equivalence in this paradigm and to show that graph-isomorphism–based testing provides a sound alternative. This represents a methodological contribution that is independent of any particular algorithmic component.
>
> **2. Theoretical novelty: identifying the SD condition and characterizing the regime where WL becomes reliable.**
>
> While the Weisfeiler–Lehman (WL) test itself is not new, the structural characterization of when WL can reliably certify equivalence for optimization models is new. We identify the Symmetric Decomposable (SD) condition and theoretically establish that WL is guaranteed to be correct under SD. This boundary characterization is crucial for adapting generic GI algorithms to the domain of optimization modeling.
>
> **3. Experimental contributions through Bench4Opt and LLM benchmarking.**
>
> We introduce Bench4Opt, a model–data separated benchmark for optimization modeling. Using this dataset, we (i) empirically demonstrate that ORGEval is efficient and consistent, and (ii) provide the first systematic benchmark of state-of-the-art LLMs on optimization modeling tasks.
>
> Together, these components reflect substantial novelty in methodology, theory, and empirical evaluation, beyond simply incorporating an SD detection in our algorithm.

---

> ### Author Response · Authors · 2025-11-28
> **In response to Question 1**
>
> > Is it possible to have non-SD instances that are equivalent? Does the framework just reject them altogether?
>
> Thanks for the question. Our framework is designed for the LLM benchmarking and training setting, where the evaluation data and training data can be selected to satisfy the SD (symmetric decomposable) condition. In practice, this is ensured by running our SD-detection algorithm on all ground-truth models before they enter the benchmark. Therefore, the application scenario never includes non-SD ground-truth instances. Given this setup, if an LLM-generated model is non-SD, the framework can safely classify it as incorrect. In other words, although non-SD but equivalent formulations might exist in theory, they do not arise in our benchmark, and the evaluation framework is intentionally designed to reject such cases.
>
> Regarding the reviewer’s broader concern might be about how common non-SD problems are in real applications, as discussed in responses to weakness 1 and question 1 from Reviewer Ntbm, empirical evidence suggests that non-SD LP/MILPs are rare.

---

> ### Author Response · Authors · 2025-11-28
> **In response to Weakness 2**
>
> > Limited testing on other datasets, only worked on its own benchmark.
>
> Thanks to the reviewer for raising this concern. We agree that broader testing is important. However, model–data separated benchmarks are extremely rare in this domain. Most existing benchmarks do not provide ground-truth models. This is why we constructed a new benchmark tailored for evaluating our proposed method.
>
> To our knowledge, Text2Zinc is the only other benchmark that also provides ground-truth models. However, its data format is not directly compatible with ours. Our algorithm(code) is currently tailored to optimization instances represented in '.lp' files. Adapting Text2Zinc to our setting, therefore, still requires model/data transformations. Nevertheless, we plan to include this adaptation and the corresponding experimental comparison in the camera-ready version.
>
> Finally, we would like to emphasize that the benchmark serves primarily as an application of the proposed ORGEval. The main contribution of our work lies in introducing a new evaluation methodology, which is orthogonal to the specific dataset used. Moreover, our benchmark leads to conclusions consistent with those observed across other benchmarks in the field.

---

> ### Author Response · Authors · 2025-11-28
> **In response to Weakness 3**
>
> > The benchmark dataset is also relatively small
>
> We would like to clarify that, within the domain of OR modeling with LLM, the scale of our benchmark dataset is well aligned with existing standards. For reference, we summarize several representative benchmarks:
>
> | Benchmark | # Problems | Ground-truth model available? |
> | -------------------- | ---------- | ----------------------------- |
> | Complex OR | 37 | No |
> | NLP4LP | 67 | No |
> | IndustryOR | 100 | No |
> | TEXT2ZINC | 110 | Yes |
> | MAMO-ComplexLP | 211 | No |
> | **Bench4OPT (ours)** | **394** | **Yes** |
> | Optibench | 605 | No |
> | MAMO-EasyLP | 640 | No |
>
> Compared with prior datasets, especially the one that include model–data separation and verified ground-truth models, Bench4OPT is substantially larger. In fact, the only existing benchmark with model-data separation prior to ours contains 110 problems, whereas Bench4OPT expands this to 394 problems.
>
> Furthermore, each problem in our benchmark has been carefully reviewed by domain experts in operations research to ensure correctness and practical relevance. This expert-level verification is crucial for reliability but inherently limits the scale of Bench4Opt

---

> ### Author Response · Authors · 2025-11-28
> **In response to Weakness 4 and 5**
>
> > Limited explanation on testing performacne on different LLMs on the benchmark. e.g.
> > Accuracy is binary (equivalent v.s. not-equivalent) - Would be interesting to see the degree to which a model is inaccurate and also how (e.g. is it a matter of unable to translate the requirements?)
>
> Thanks for the suggestion. We agree that understanding how models fail is essential for interpreting benchmark performance and diagnosing model limitations.
>
> Due to time constraints, we conducted an additional error analysis on three representative LLMs (GPT-4o, GPT-5, and O1). We annotated 1181 answers and categorized them into two major groups: **compile errors** and **modeling errors**.
>
> Compile Errors include issues visible directly from the execution logs, including
>
> - Basic Python mistakes: indentation errors, syntax errors, etc.
>
> - Gurobi-specific errors: incorrect function names, etc.
>
> - Code errors stemming from a misunderstanding of the OR model: e.g., misunderstanding input formats or the meaning of variables.
>
> Modeling Errors  include mistakes in the optimization formulation:
>
> - Incorrect objective functions
>
> - Incorrect constraints
>
> - Incorrect decision variables
>
> Importantly, modeling errors may co-occur, so we report their full incidence. The distribution of error types (percentage of test errors attributable to each type) is:
>
> | Error Type | GPT-4o | GPT-5 | O1 |
> | ---------------------- | ------ | ----- | ----- |
> | Python understanding | 9.97 | 0.41 | 0.00 |
> | Gurobi understanding | 5.30 | 0.41 | 0.40 |
> | Modeling understanding | 4.36 | 2.48 | 1.61 |
> | Wrong objective | 3.43 | 2.07 | 3.23 |
> | Wrong constraints | 56.07 | 77.69 | 60.08 |
> | Wrong variable | 27.41 | 32.23 | 39.92 |
>
> **Key Observations.**
>
> 1. As LLMs improve, compile errors related to basic coding abilities (Python/Gurobi) steadily decrease (O1 < GPT-5 < GPT-4o).
>
> 2. constraint-related mistakes dominate overall failures.
>
> Due to limited time, we completed this analysis for only three LLMs. However, we fully agree that broader coverage would significantly enhance the benchmark’s diagnostic value. We plan to extend this error-type analysis to more LLMs in the camera-ready version.

---

> ### Author Response · Authors · 2025-11-28
> **In response to Question 3**
>
> Thank you for the question. Our benchmark consists of 394 optimization problems, including 200 LP instances and 194 MILP instances. Among them, 340 instances are generated from LLM-proposed seed problems that were subsequently verified by domain experts, and 54 instances are generated from seed problems selected from MIPLIB.
>
> Regarding a per-instance breakdown, many of the generated models exhibit overlapping structural characteristics, and there is no universally accepted taxonomy for mapping real-world optimization formulations to a single problem class. For the same reason, many application scenarios are inherently multi-label. As a result, forcing a strict instance-by-instance categorization would be somewhat arbitrary and potentially misleading. To provide clarity while avoiding such artifacts, we instead report a set of problem classes and application domains covered by Bench4Opt in **Appendix B.3**.

---

### Official Review · Reviewer_eNh4 · 2025-11-04

**Soundness:** 3
**Presentation:** 2
**Contribution:** 4
**Rating:** 6
**Confidence:** 4

**Summary:**

In this paper, the authors introduced a new metric to evaluate the accuracy of formulating mixed-integer linear programs (MILPs) by large language models (LLMs). Specifically, instead of checking the optimal value of the formulated MILP, the authors proposed to check the "structural equivalence", measured by isomorphism of the underlying graphs of the MILPs. The new metric fixes several limitations of the common practice of checking the optimal values. The author also proposed an algorithm to perform the equivalence check.

Overall, I think this paper addresses an important issue of performance metrics in auto-formulation through an innovative approach.
However, I do have some questions in terms of the clarity of the definitions, the experimental results, and the implementation details.

**Strengths:**

* I do agree that there are flaws in checking optimal values to determine the correctness of problem formulation. To my knowledge, this paper is the first to address this important issue.
* The authors attempt to rigorously define the model equivalence and various related concepts.
  * Some definitions need clarification, though. Please see my comments in "Weaknesses".
* The authors constructed a new benchmark dataset to check formulation correctness based on the proposed model equivalence metric.
  * It would be great if the authors can make the new benchmark publicly available.

**Weaknesses:**

**Soundness and Clarity of Various Definitions.** The authors made several definitions in terms of model $\mathcal{M} (`Definition 2`), model isomorphism (`Definition 5`), and graph (`Definition 7`). It is important to make these definitions consistent and coherent. Below are some comments and questions.
* It is not clear what is the exact definition of *formulation correctness*, namely the equivalence between the ground-truth formulation and the formulation given by a LLM. If I understand correctly, a "model" is represented by the underlying graphs $\mathbf{G}=(\mathbf{V} \cup \mathbf{W}, \mathbf{E})$ (`Definition 7`), and the "problem data" consists of the weights of edges $\mathbf{A}$ and features of vertices $(\mathbf{b}, \mathbf{c}, \circ, \tau)$. In which of the following scenarios do we say the LLM "correctly formulates" the problem?
  * The "model", represented the graph $\mathbf{G}$, is correct;
  * *Both* the "model" $\mathbf{G}$ *and* the "problem data" $(\mathbf{A}, \mathbf{b}, \mathbf{c}, \circ, \tau)$ are correct; or
  * The "model" $\mathbf{G}$ is correct, and *any randomly sampled* "problem data" is correct (this seems to be consistent with `Definition 5`).
* I think it is also important to clarify what happens when $A_{ij}=0$ in the problem description (e.g., a constraint like $2 x_1 + 4 x_3 \leq 0$, where $x_2$ is not involved). What is the underlying graph? Is it a graph with *no edge* between $i$ and $j$, or a graph with a *zero-weight edge* between $i$ and $j$? If both graphs are eligible, are they isomorphic?
* The algebraic definition of the "model" and "problem data"/"modeling parameter" in `Definition 2` may not be consistent with the graphical definition in `Definition 7`. In `Definition 2`, the problem data seems to be any tuple $\theta=(\mathbf{A}, \mathbf{b}, \mathbf{c}, \circ, \tau)$ even with different dimensions $m,n,p$ (as illustrated in `Figure 8`). So different problem data can result in different graphs (i.e., different numbers of nodes and topologies). Then *all* MILPs can be represented by *one* model $\mathcal{M}$ with different $\theta$. This seems unreasonable to me.
* In `Definition 3` of model-lossless-reduction, what are the restrictions of the mapping $F$? Does the same mapping $F$ exist for *any* parameter $\theta$? Could you please provide an example?

**Algorithm and Implementation.** From `Definition 5` and `Definition 6`, it seems that we mainly need to check if the LLM formulation is the same as the ground-truth formulation, subject to *permutations of variables and constraints*.
* If it is just permutations of variables and constraints, are there easier methods than checking graph isomorphism?
  * For example, [Astorga et al., 2025](https://openreview.net/forum?id=33YrT1j0O0) used *Satisfiability Modulo Theories* (SMT) to check equivalence of expressions, which goes beyond permutation equivalence.
* In implementation, how do we get the underlying graph of the LLM formulation? Did you consider the errors in translating the formulation to graphs?

**Experimental Results.** I cannot find any explanations on `Table 2` and `Table 3`.
* It is not clear what are "feasibility consistency", "ORGEval consistency", "solver consistency" in `Table 2`.
* It is not clear what are "compile error", "structured", and "unstructured" in `Table 3`.

**Questions:**

Please see my comments in "Weaknesses".

Additional question:
* What are the prompts in `Figure 4` and `Figure 5`? They contain the problem description, but also give away the answers (i.e., parameters, decision variables, objectives, constraints).

Minor typos:
1. Line 307: "see figure Figure 3".
2. Line 401: "Specifically Each Bench4Opt problem".
3. Line 405: "algorithm Algorithm 3"

---

> ### Author Response · Authors · 2025-11-28
> **In response to Weakness 1**
>
> >  In which of the following scenarios do we say the LLM "correctly formulates" the problem?
>
> We first clarify the relationship among the three concepts: *model*, *model instance*, and *graph representation*.
> - A *model* maps admissible problem data $\theta \in \Theta$ to a specific *model instance* $\mathcal{M}(\theta)$.
> - A *model instance* $\mathcal{P}$ is an initiantiation of a *model* with specific problem data, i.e.  $\mathcal{P}=\mathcal{M}(\theta)$.
> - The *graph* introduced in Definition 7 is a representation of a *model instance*, not of the *model* itself.
>
> With this hierarchy, Definition 1 (model instance), Definition 2 (model), and Definition 7 (graph representation) are fully aligned.
>
> Given this context, **none of the three proposed scenarios precisely captures what we mean by a “correct formulation”.** Our criterion is as follows:
>
> We say that an LLM correctly formulates a problem if, for any randomly sampled problem data $\theta$, the graph of the model instance produced by the LLM is isomorphic to the graph of the ground-truth model instantiated with the same $\theta$. Formally, if instances are represented in graphs, the requirement is:
> $$
> \forall \theta \in \Theta,\quad \text{Graph}(M_1(\theta)) \cong \text{Graph}(M_2(\theta)).
> $$
> This definition is consistent with Definition 5 and accurately reflects the notion of structural correctness that our benchmark evaluates.
>
>
> > What is the underlying graph when $A_{ij} = 0$? Is it a graph with no edge between $i$ and $j$, or a graph with a zero-weight edge between $i$ and $j$? If both graphs are eligible, are they isomorphic?
>
> **When** $A_{ij} = 0$**, the graph has a zero-weight edge between** $i$ **and** $j$.
> In our implementation, we do not distinguish "no edge" and "zero-weight edge" in the graph. The graph is stored using
> - A node feature matrix,
> - A constraint feature matrix,
> - An adjacency matrix $A_{adj}$, where $A_{adj}[i,j]$ encodes the edge information between node $i$ and node $j$.
>
> By definition in our representation, $A_{adj}[i,j]=0$ means there is no connection or zero connection between node $i$ and node $j$; and positive (or non-zero) entries encode actual coefficients from the MILP formulation. **Therefore,  we do not distinguish between "the edge does not exist" and "the edge has zero weight" at the graph level.**
>
> Despite the above clarification, we think a deeper concern of the reviewer might be the following:
>
> If we have two models that only differ in their constraints:
> - Model 1 with constraint 1: $a_1*x_1+a_2\*x_2 \leq b, a_2 = 0$,
> - Model 2 with constraint 2: $a_1*x_1 \leq b, (\text{no parameter } a_2)$,
>
> Would we regard them as equivalent or not?
>
> **Though the graph for these two model instances would be identical, and the two graphs are equivalent, model 1 and model 2 are structurally non-equivalent** because:
> Model 1 may allow for choosing different $a_1$ and $a_2$ to form different instances, while Model 2 only allows for choosing $a_1$.
>
> The key is that two models have different parameter support, so they can be judged as inequivalent when plugging a shared problem data before graph isomorphism testing (at least one of the models will fail to instantiate).
>
> > The algebraic definition of the "model" and "problem data"/"modeling parameter" in Definition 2 may not be consistent with the graphical definition in Definition 7
>
> Definition 2 specifies the definition of a model, which maps data to a model instance. In contrast, Definition 7 introduces the graph representation of a model instance, not of the model itself. These two definitions refer to different concepts.
>
> We clarify that Definition 1 describes a model instance, and this notion is what corresponds directly to the graph representation in Definition 7. Therefore, Definition 2 (model), Definition 1 (model instance), and Definition 7 (graph of a model instance) are consistent once this hierarchy is made explicit.
>
> > In Definition 2, the problem data seems to be any tuple $\theta=(\mathbf{A}, \mathbf{b}, \mathbf{c}, \circ, \tau)$ even with different dimensions $m,n,p$ (as illustrated in Figure 8). So different problem data can result in different graphs (i.e., different numbers of nodes and topologies).
>
> Different data can result in different graphs, that's true because a graph is a representation for a model instance, and a model instance is instantiated by a fixed tuple $\theta=(\mathbf{A}, \mathbf{b}, \mathbf{c}, \circ, \tau)$.

---

> ### Author Response · Authors · 2025-11-28
> **In response to  Weakness 1 - continue**
>
> > all MILPs can be represented by one model $\mathcal{M}$ with different $\theta$. This seems unreasonable to me.
>
>
> Thanks to the reviewer for raising this insightful issue. Under the previous formulation, a “model” was defined as a mapping $\theta \mapsto \mathcal{P},$ where $\theta$ belongs to a model-specific parameter space $\Theta$. As the reviewer observes, if $\Theta$ were unconstrained, then in principle all MILPs could be represented by a single model $\mathcal{M}$ equipped with different choices of $\theta \in \Theta$. **This was not our intention. The key point is that, in our setting, $\Theta$ is not arbitrary (see Appendix E, Example 4).**
>
> We also realized that this formulation may be misleading, because under that definition the notion of a “model’’ is determined almost entirely by the model-specific parameter space $\Theta$ for the mapping, rather than by the mapping itself. To clarify this, **we have revised the definition as follows：**
>
> For fixed dimensions $p$, a model $\mathcal{M}$ is a mapping： $\mathcal{M}: \Theta \subset \mathbb{R}^p \to \mathcal{P}$, where $\Theta$ is the problem data support, and $\mathcal{P}$ is a problem instance, instantiated with $(A,b,c,\circ,\tau)$. Here, $\Theta$ may be any subset of $\mathbb{R}^p$ that represents the admissible “data" of the modeling. In practice, part of $\Theta$ is fixed (structural components), and the remaining part consists of real-valued elements that vary within a closed subset of $\mathbb{R}$.
>
> Regarding the example in Figure 8, we further realized that not every model can be instantiated *at arbitrary dimensions* $(m,n,p)$. For this reason, we removed this example in the revision. This does not affect our theoretical results. (In implementation, we can indeed generate data of varying dimensions—each model has corresponding data-generation code that serves as a sampler to sample from $\Theta$. However, for conceptual clarity in the theory, we focus on fixed dimensions.)
> **We have revised the definitions accordingly in the updated manuscript. We sincerely thank the reviewer, as this comment helped us improve the clarity and correctness of our framework.**
>
> > In Definition 3 of model-lossless-reduction, what are the restrictions of the mapping $F$? Does the same mapping $F$ exist for any parameter $\theta$? Could you please provide an example?
>
> The definition **does not impose structural restrictions on the mapping** $F$; it only requires that $F$ be solution-preserving for all admissible parameters. For some models, such an $F$ does exist uniformly for every $\theta \in \Theta$. For example, consider the following problem：
>
> A farmer who seeks to maximize profit using T acres of land to grow wheat and rice. Wheat yields $M_1$ kilograms per acre and rice yields $M_2$ kilograms per acre; each kilogram of wheat brings a profit of $C_1$, and each kilogram of rice brings a profit of $C_2$.
> - Problem data support: $(T,M_1,M_2,C_1,C_2) \subset \mathbb{R}_+^5$;
> - Model1: $\mathcal{M}_1: (T,M_1,M_2,C_1,C_2) \to \min -M_1\*C_1\*x_1 - M_2\*C_2\*x_2;s.t., x_1 + x_2 \leq T;x_1, x_2 ≥ 0$;
> - Model2: $\mathcal{M}_2: (T,M_1,M_2,C_1,C_2) \to \max M_1\*C_1\*y_2 + M_2\*C_2\*y_1; s.t., y_2 <= T-y_1;y_2, y_1 ≥ 0$.
>
> In this case, $F$ maps $(x_1*,x_2*)  \text{ to } (y_2*,y_1*)$, and this mapping is just a permutation and reindexing of decision variables, it is independent of  $\theta$.

---

> ### Author Response · Authors · 2025-11-28
> **In response to Weakness 2.1**
>
> >  In response to "If it is just permutations of variables and constraints, are there easier methods than checking graph isomorphism?" and comparison with SMT.
>
> Thanks the reviewer for introducing SMT, we believe this is a highly related method for modeling equivalence evaluation and we are happy to discuss this and cited https://openreview.net/forum?id=33YrT1j0O0 in our paper (see our revised manuscript introduction section.) **We believe the question touches on two related but conceptually different tasks**, and we clarify them below.
>
> - Permutations of variables and constraints.
>
> To check permutation equivalence between model instances，one will identify a permutation of variables and constraints that transforms model instance $A$ into model instance $B$. In the worst case, a naive algorithm would need to consider all possible permutations of the node set, which is factorial in size. This problem is equivalent to a graph-isomorphism (GI) testing problem(see our Lemma D.1). **Any general method for recovering such a permutation must, in effect, solve a GI problem, so there is no simpler universal alternative than graph-isomorphism testing.**
>
> - SMT-based equivalence.
>
> We would first like to acknowledge the strength of SMT solvers in checking logical equivalence between model formulations. SMT is highly effective for verifying equivalence when the correspondence between variables is known.
> In our setting, however, the challenge is different. **SMT-based equivalence checking does not address the permutation problem: an SMT solver can verify whether two formulas are equivalent only after the correspondence between decision variables has been explicitly specified.** When variables are permuted and no correspondence is provided, standard SMT formulations will report the two models as inequivalent, even when they are identical up to relabeling.
>
> We include in the anonymous link an example where two optimization models are equivalent under a simple permutation, yet the SMT solver returns “not equivalent”. （https://anonymous.4open.science/r/ORGEval-Rebuttal-ICLR2026-009D/example_smt_fail/model_reference_smt.lp  vs https://anonymous.4open.science/r/ORGEval-Rebuttal-ICLR2026-009D/example_smt_fail/model_test_smt.lp ）
>
> In this sense, SMT implicitly assumes away one component of model equivalence: discovering the permutation that aligns two models structurally. This is precisely the aspect that graph-isomorphism-based approaches are designed to handle explicitly.
>
> **One might respond that this issue can be resolved by finding the mapping first. However, this is one of the hardest parts for model equivalence evaluation.** https://openreview.net/forum?id=33YrT1j0O0 and EquivaMap(https://arxiv.org/pdf/2502.14760) attempt to generate variable mappings using LLMs. But this approach has two major weaknesses:
> - Producing a correct mapping requires calling the LLM with the full formulations, sometimes repeatedly. This is significantly more expensive than structural methods that operate directly on the model representation.
> - LLMs frequently produce incomplete or incorrect correspondences—especially for formulations with many indexed variables, auxiliary constructs, or redundant constraints. Any such error immediately causes the subsequent SMT verification to fail, even when the two formulations are truly equivalent.
>
> Thus, **finding the permutation mapping is a nontrivial inference problem that SMT assumes away and that LLM-based mapping methods cannot reliably handle.** GI-based and structural-evaluation approaches address this issue directly by comparing the formulations at the structural level, without depending on solvers or on LLM-generated mappings.

---

> ### Author Response · Authors · 2025-11-28
> **In response to Weakness 2.2**
>
> > In implementation, how do we get the underlying graph of the LLM formulation?
>
> Thanks for the reviewer's interest in our implementation details. We first parse the .lp file produced by compiling the LLM-generated modeling code and extract all structural information through a solver interface (e.g., Gurobi). This gives us:
> - the list of decision variables, including their types (integer / continuous) and objective coefficients;
> - the full set of constraints, including constraint sence $\(\leq, \geq, =\)$ and right-hand-side constants;
> - the coefficient matrix showing the weight of each variable in each constraint.
>
> Using this information, we construct a bipartite graph. The graph representation consists of:
>
> - Adjacency matrix $A_{adj}$:
>  Each entry $A_{adj}$  stores the coefficient connecting decision-variable node $i$ and constraint node $j$.
>  If two nodes are not connected, then $A_{adj}[i,j] = 0$.
> - Decision-variable node feature matrix, each row corresponds to one variable and includes the following features:
>     - variable’s objective coefficient,
>     - variable type (integer / continuous).
> - Constraint node feature matrix, each row corresponds to one constraint and includes the following features:
>     - the constraint sense $(\leq, \geq, =)$,
>     - The right-hand-side constant coefficient.
>
> Thus, each **LP/MILP formulation is translated into a triplet: (Adjacency matrix, Variable feature matrix, Constraint feature matrix), and this triplet characterizes a graph.**
>
> > Did you consider the errors in translating the formulation to graphs?
>
> Thanks for raising this question. We did encounter a translation error. In practice, variable bounds can be both modeled as `x.lb`/`x.up` in GUROBI or appear as explicit constraints of the form $x \circ b$, even though they are mathematically identical. Without correction, these differences would lead to non-isomorphic graphs for equivalent formulations.
>
> To avoid this, before graph construction, we perform an LP normalization step: all variable bounds are converted into standard one-sided linear constraints. Only after normalization do we translate the LP into graph form. This ensures that two semantically equivalent models produce consistent graph representations and prevents false mismatches caused by formulation formatting artifacts.

---

> ### Author Response · Authors · 2025-11-28
> **In response to Weakness 3 and Question**
>
> Thanks for pointing out this clarity issue. We have revised the paper to better define these terms.
>
> ## Experiment in table 2
>
> The experiment in Table 2 aims to illustrate the following question:
>
> If we use the evaluation result in one model instance, to what extent can we make the same conclusion on another model instance under the same model?
>
> For each Bench4OPT problem, we evaluate LLM-generated formulations using both the Solver-based method and ORGEval over 5 randomly instantiated data instances. The table reports the proportion of instances for which the two evaluation schemes yield consistent results. We represent the following 3 metrics:
>
> - **ORGEval consistency** denotes the percentage of problems whose five instances have a consistent evaluation result under ORGEval.
> - **Solver consistency** denotes the percentage of problems whose five instances have consistent evaluation results under solver-based optimal value matching.
> - **Feasibility consistency** denotes the percentage of problems whose five instances can all be solved by the solver.
>
> ## Experiment in table 3
>
> The experiment in Table 3 aims to benchmark several LLMs on optimization modeling. We prompt each LLM with modeling problems in Bench4OPT once and use ORGEval to evaluate LLM-generated model against the corresponding ground-truth model. The evaluation metric includes:
>
> - **Compile error**: In our experiment, we ask LLM to build the model and write it in GUROBI code. Therefore, compile error is the percentage of problems for which the LLM-generated Gurobi code fails to compile/run.
> - **Accuracy**: the percentage of problems whose LLM-generated model is isomorphic to the corresponding ground-truth model.
>
> In the Bench4Opt dataset, each problem can be represented at two levels of abstraction: a structured level and an unstructured level.
>
> - **Structured**:Following the conventions of INFORMS optimization modeling competitions, we design a problem template in which each instance contains explicit and well-organized sections, including background, problem description, parameters, objective, constraints/decision variables, implementation notes, and expected output format.
> - **Unstructured**: For this version, we merge and rewrite all descriptive sections (except the explicit parameter descriptions) into one paragraph. Compared with the structured setting, this version removes the information hierarchy and formatting cues, making the modeling task more complex.
>
> Figures 4 and 5 are examples of structured and unstructured problem descriptions. We updated the descriptions in Table 2, Table 3, Figure 4, and Figure 5. We also corrected several typos.

---

### Author Response · Authors · 2025-12-03
**Summary of Rebuttal**

We would like to thank the reviewers for recognizing our effort and contribution in the following aspects:
- **Novelty:** We are the first to 1) address the limitation of solver-based evaluation for model-equivalency (Reviewer `eNh4`), and 2) use graph to evaluate model equivalence (Reviewer `WWVn` and `NtBm`)
- **Theoretical contribution:** We rigorously defined model equivalence (Reviewer `eNh4`) and characterized a sufficient condition of graph isomorphism testing (Reviewer `NuHH` and `kxC7`)
- **Benchmark dataset:** We propose a model-data separated benchmarking dataset, Bench4Opt (Reviewers `eNh4`, `NtBm`, `kxC7`), which is recognized as an important and valuable contribution to the field
- **Empirical Strengths:** Our proposed evaluation method, ORGEval, achieved empirical advantage in efficiency and consistency (Reviewer `NuHH`).

---

Based on the reviewers' questions, we made the following clarifications:

## Scope of our study
The “major weakness” identified by Reviewer `kxC7`’ arises from a misunderstanding of our scope. The reviewer interprets our method as an online verification tool and therefore argues that verifying LLM-generated models remains fundamentally limited by access to ground truth. To address the misunderstanding of reviewer `kxC7`, we clarified our scope as follows:
- Our goal is to provide a principled benchmarking framework for evaluating LLMs’ capability in optimization-model translation, rather than building an online verification tool for real-world deployment. ORGEval assumes access to a reference ground-truth formulation, which is standard in benchmarking. The main motivations for this work are emphasized in the rebuttal:
	- Benchmarking is essential for scientific progress in automatic modeling.
	- Our work extends the existing benchmarking paradigm toward full structural evaluation, providing a more informative and stable correctness signal than optimal-value comparison.

Reviewers `WWVn` and `NtBm` raise concerns in distinguishing problems that always have the same optimal solution but are structurally different. We clarified that our evaluation method is designed for benchmarking a "natural-language to model" translation task, which strictly assesses the semantic equivalence between two models.
- Our evaluation focuses on detecting whether an LLM can correctly translate a textual OR problem description into a mathematical formulation. Under this benchmarking objective, we regard the modeling task as a specialized translation task, and a formulation is considered correct only if it strictly adheres to the textual description.
- More detailed discussion could be found in the response for Weakness 1 and 2 from Reviewer `WWVn`, and Weakness 3 from Reviewer `NtBm`.

## Clarification on problem setting and definitions

Reviewer `eNh4` raises concerns about the consistency of the definitions of model, model instance, and graph representation. To address these concerns, we have added clarifications that make the relationships among these concepts more explicit.

- A model maps problem data to a model instance, and a graph defined in Definition 7 represents only the model instance, not the model itself. In response to reviewer eNh4 's question, we refine our definition of a “model”, clarifying that its admissible data space is not arbitrary, preventing the unintended implication that all MILPs belong to a single model.
 - Detailed discussion in problem setting and definitions can be found in our response to Reviewer `eNh4` Weakness 1

## Coverage of our theoretical guaranteed sufficient condition (SD) for GI

Both Reviewer `NtBm` and Reviewer `kxC7` raised concern regarding the coverage of SD We clarified this through additional experiments and additional realistic examples:

- The SD condition we characterized is both useful and commonly satisfied in optimization-modeling scenarios, providing the following evidence:
	- The SD condition holds widely in practice because for many problems, random parameter assignments naturally break graph symmetries, making common optimization models decompose into non-symmetric subgraphs and thereby simplifying isomorphism detection.
	- We validate the SD property on a broad set of problems.
		- All LLM-generated classic problems with random parameterization are SD.
		- All instances with a reasonable size in the symmetric TSPLib dataset are SD.
	- We provided several realistic SD examples in Appendix E.3 for better understanding.
- More detailed discussion is in "response to Weakness 1 and Question 1" for Reviewer `NtBm`; and "On Graph Isomorphism Testing and SD condition - continue" for Reviewer `kxC7`.

 ## Open-source code and benchmark dataset:

Reviewers `eNh4` and `NtBm` requested that the data be made publicly available. We clarified that

- Our benchmark is already included in the Supplementary Material (in the zip folder) on OpenReview in the original submission.

---

> ### Author Response · Authors · 2025-12-03
> **Summary of Rebuttal - continue**
>
> According to reviewers' suggestions, we also made the following adjustments in our revised manuscript
> - Adjustments on the problem setting part to improve clarity. （Section 2.1.1）
> - More discussion on related works (Section 1, Appendix C)
> - Error analysis depicting the performance difference between the reasoning model (o1) and base model (gpt-4o, gpt-5) (Appendix F)
> - Data construction detail and dataset breakdown (Appendix B)
> - More detailed explanations about Table 2 and Table 3.

---

### Note · Program_Chairs · 2026-01-17
**Submission Desk Rejected by Program Chairs**

The following references in this submission do not refer to real documents and/or have major errors in bibliographic information:

 Team ApIO, Santiago Ramírez Palacio, Mariana Escallón Barrios, and Daniel López Cornejo. 9th aimms-mopta optimization modeling competition (2017) production and delivery of radiopharmaceuticals to medical imaging centers.